# Obesity-associated NLRC4 inflammasome activation drives breast cancer progression

Ryan Kolb[1,2], Liem Phan[3,4], Nicholas Borcherding[1,5], Yinghong Liu[6], Fang Yuan[6], Ann M. Janowski[7], Qing Xie[1,8], Kathleen R. Markan[8,9], Wei Li[1], Matthew J. Potthoff[8,9], Enrique Fuentes-Mattei[10], Lesley G. Ellies[11], C. Michael Knudson[1], Mong-Hong Lee[3,4,12], Sai-Ching J. Yeung[3,13,14], Suzanne L. Cassel[7,15,†], Fayyaz S. Sutterwala[7,15,†] & Weizhou Zhang[1,5,7,16,17]

Obesity is associated with an increased risk of developing breast cancer and is also associated with worse clinical prognosis. The mechanistic link between obesity and breast cancer progression remains unclear, and there has been no development of specific treatments to improve the outcome of obese cancer patients. Here we show that obesity-associated NLRC4 inflammasome activation/ interleukin (IL)-1 signalling promotes breast cancer progression. The tumour microenvironment in the context of obesity induces an increase in tumour-infiltrating myeloid cells with an activated NLRC4 inflammasome that in turn activates IL-1β, which drives disease progression through adipocyte-mediated vascular endothelial growth factor A (VEGFA) expression and angiogenesis. Further studies show that treatment of mice with metformin inhibits obesity-associated tumour progression associated with a marked decrease in angiogenesis. This report provides a causal mechanism by which obesity promotes breast cancer progression and lays out a foundation to block NLRC4 inflammasome activation or IL-1β signalling transduction that may be useful for the treatment of obese cancer patients.

[1] Department of Pathology, University of Iowa Carver College of Medicine, Iowa City, Iowa 52242, USA. [2] Center for Immunology and Immune Based Diseases, University of Iowa Carver College of Medicine, Iowa City, Iowa 52242, USA. [3] University of Texas Graduate School of Biomedical Sciences at Houston, Houston, Texas 77030, USA. [4] Department of Molecular and Cellular Oncology, University of Texas MD Anderson Cancer Center, Houston, Texas 77030, USA. [5] Medical Scientist Training Program, University of Iowa Carver College of Medicine, Iowa City, Iowa 52242, USA. [6] Department of Nephrology, The Second Xiangya Hospital, Research Institute of Nephrology, Central South University, Changsha, Hunan 410011, China. [7] Interdisciplinary Graduate Program in Immunology, University of Iowa Carver College of Medicine, Iowa City, Iowa 52242, USA. [8] Department of Pharmacology, University of Iowa Carver College of Medicine, Iowa City, Iowa 52242, USA. [9] Fraternal Order of Eagles Diabetes Research Center, University of Iowa Carver College of Medicine, Iowa City, Iowa 52242, USA. [10] Department of Experimental Therapeutics, The University of Texas MD Anderson Cancer Center, Houston, Texas 77054, USA. [11] Department of Pathology, University of California, San Diego, La Jolla, California 92093, USA. [12] Cancer Biology Graduate Program, University of Texas MD Anderson Cancer Center, Houston, Texas 77030, USA. [13] Department of Emergency Medicine, University of Texas MD Anderson Cancer Center, Houston, Texas 77030, USA. [14] Department of Endocrine Neoplasia and Hormonal Disorders, University of Texas MD Anderson Cancer Center, Houston, Texas 77030, USA. [15] Department of Internal Medicine, University of Iowa Carver College of Medicine, Iowa City, Iowa 52242, USA. [16] Free Radical and Radiation Biology Program, University of Iowa Carver College of Medicine, Iowa City, Iowa 52242, USA. [17] Cancer Genes and Pathway Holden Comprehensive Cancer Center, University of Iowa Carver College of Medicine, Iowa City, Iowa 52242, USA. † Present address: Department of Medicine, Cedars-Sinai Medical Center, Los Angeles, California 90048, USA. Correspondence and requests for materials should be addressed to W.Z. (email: weizhou-zhang@uiowa.edu).

Obesity has an impact on 36% of US adults, and is attributed to ∼16–20% of cancer deaths in women and 14% of cancer deaths in men[1]. Obesity is not only associated with an increased risk of oestrogen receptor (ER)-positive breast cancer in postmenopausal women, but also is associated with a worse clinical outcome independent of menopausal status[2–4]. Many factors have been proposed for obesity-driven breast-cancer progression (ODBP), including oestrogen, insulin resistance, the balance between adipokines, as well as pro-inflammatory cytokines including IL-6 and tumour necrosis factor-α[5]. However, the direct evidence for the requirement of these factors in ODBP is largely missing. Of particular interest in obesity is its association with chronic inflammation[6]. Obesity and chronic inflammation are known risk factors for many chronic conditions including diabetes, cardiovascular disease and various cancers[7–9]. Whether obesity has an impact on these diseases through enhancement of inflammation or via a direct mechanism is largely unclear; however, obese adipose tissue is a reservoir for macrophages with activated inflammasomes that contribute to insulin resistance and diabetes[10,11]. Inflammasomes are a group of multiprotein complexes comprising NOD-like receptors (NLRP1, NLRP3 and NLRC4) and Pyrin member AIM2, the adaptor protein ASC and caspase 1 (CASP1), the latter being the common effector enzyme that cleaves pro-IL-1β and pro-IL-18 to their active secreted forms[12]. While there has been no reported causal role for inflammasomes and the IL-1/IL-1R axis in breast cancer progression, IL-1β level is higher in invasive ductal carcinoma compared with ductal carcinoma *in situ* (non-invasive) and associated with a more aggressive phenotype[13]. Arendt *et al.*[14] identified the involvement of macrophages in mediating obesity-associated angiogenesis in humanized mammary adipose tissue via CCL2 and IL-1β.

In this study we identified a novel causal link between obesity and breast cancer progression, that is, via the activation of NLRC4 inflammasome and the consequent IL-1 activation in tumour-infiltrating macrophages and its downstream activation of angiogenesis via upregulation of VEGFA in adipocytes.

## Results

**Obesity is associated with IL-1 signalling in breast cancer.** To examine the link between obesity (body mass index (BMI) greater than 30) and inflammation in breast cancer, we first looked at obesity-associated changes at the transcriptomic level in human ER$^+$ breast cancers using our published data set[15]. We found that obesity is associated with an increase in several inflammatory pathways (Supplementary Fig. 1a), including significant upregulation of the IL-1 pathway (Supplementary Fig. 1b, $P < 0.05$). To further look at a potential role of the IL-1 pathway in ODBP, we also compared the transcriptomes between tumours formed from normal or obese MMTV-*TGFα* mice, a model for ER$^+$ breast cancer in which obesity promotes disease progression[15]. We found that tumours from obese mice also had a significantly elevated IL-1 signalling pathway (Supplementary Fig. 1c, $P < 0.05$, Fisher's exact *t*-test). We analysed the mRNA expression of different genes and identified a consistent increase in several genes within the IL-1 pathway from cancer specimens of obese patients (Supplementary Fig. 1d, left panel) or obese mice (Supplementary Fig. 1d, right panel) related to those from the corresponding normal-weight controls.

**Diet-induced obesity promotes tumour growth.** To further examine the link between obesity, pro-inflammatory pathways and breast cancer progression, we used two syngeneic orthotopic transplant models in obesity-prone C57BL/6 mice. Py8119 cells

are derived from an *MMTV-PyMT* mouse on the C57BL/6 background and are reported to be ER-negative[16]. We found that Py8119 tumours have greater than 40% Ki-67-positive staining (Supplementary Fig. 2a). E0771 cells are derived from a spontaneous breast adenocarcinoma in C57BL/6 mice[17], and are characterized as luminal breast cancer[18]. We found that ∼50% of tumour cells are Ki-67-positive (Supplementary Fig. 2a), indicative of luminal B type. Six-week-old C57BL/6 female mice were fed with high-fat diet (HFD), nutrient-controlled normal diet (Cont ND, containing the same macronutrients as the HFD except the fat content) or regular normal diet (ND) provided by our vivarium for 10 weeks *ad libitum* (Supplementary Fig. 2b). HFD-fed mice had increased body weight compared with both ND groups (Supplementary Fig. 2c,d). Both Py8119 and E0771 tumour growth was significantly higher in diet-induced obese (DIO) mice (Supplementary Fig. 2e,g). As we found no significant difference between two ND groups (Supplementary Fig. 2f), we chose to use the regular ND. Linear regression analysis of tumour volume versus body weight indicated a strong correlation between body weight and tumour burden (Supplementary Fig. 2h). While in patients, obesity is associated with a worse clinical outcome and increased incidence of more aggressive triple-negative breast cancer regardless of menopausal status, these effects of obesity are exacerbated in postmenopausal women[2]. We used ovariectomized (OVX) mice to mimic postmenopausal conditions and found that DIO–OVX mice had significantly increased body weight (Supplementary Fig. 2i) and tumours (Supplementary Fig. 2j) relative to DIO mice without OVX. These data validate the two models to study the effects of ODBP.

**The IL-1β/IL-1R1 axis is required for ODBP.** To determine the function of IL-1 signalling in tumour growth, mice were treated intraperitoneally (i.p.) with either IL-1R1-blocking antibody (anti-IL-1R1)[19] or isotype IgG. We verified the efficacy of anti-IL-1R1 *in vivo* by measuring the relative expression of IL-1-responsive genes and saw a significant reduction in the expression of *Il6*, *Il1b* and *Ptgs2* in tumours from anti-IL-1R1-treated mice relative to IgG-treated mice (Supplementary Fig. 2k). Anti-IL-1R1-treated DIO mice had significantly reduced tumour growth compared with control IgG-treated DIO mice or non-treated DIO mice (Fig. 1a). Interestingly, the tumour growth of anti-IL-1R1-treated DIO mice was similar to that of non-treated ND mice (Fig. 1a), while there was no difference in tumour growth in ND mice treated with anti-IL-1R1 or control IgG (Supplementary Fig. 2l). We also treated tumour-bearing DIO mice with anakinra, a recombinant IL-1 receptor antagonist approved by FDA to treat other human diseases, and found a similar result as with anti-IL-1R1 antibody (Fig. 1b). Blocking IL-1R1 had the same effect on E0771 tumour growth in DIO mice (Fig. 1c). IL-1R1 blockage interrupted signalling by IL-1α and IL-1β. $Il1α^{-/-}$ mice had similar weight gain as wild-type (WT) mice when fed HFD diet (Fig. 1d), showing no significant difference in tumour growth relative to their WT counterparts, nor did they when fed the ND compared with WT mice fed a ND (Fig. 1e). $Il1β^{-/-}$ mice, however, failed to gain weight with HFD (Fig. 1f) and had reduced tumour growth in ND- and HFD-treated mice, compared with the WT counterparts (Fig. 1g). We thus conclude that the IL-1β/IL-1R axis is required for ODBP.

**NLRC4 inflammasome is required for ODBP.** While there are a number of inflammasomes that are potentially responsible for IL-1β production, adipose tissue-infiltrating macrophages have activated NLRP3 inflammasome[10]. We had thus reasoned that the NLRP3 inflammasome is required for IL-1β production

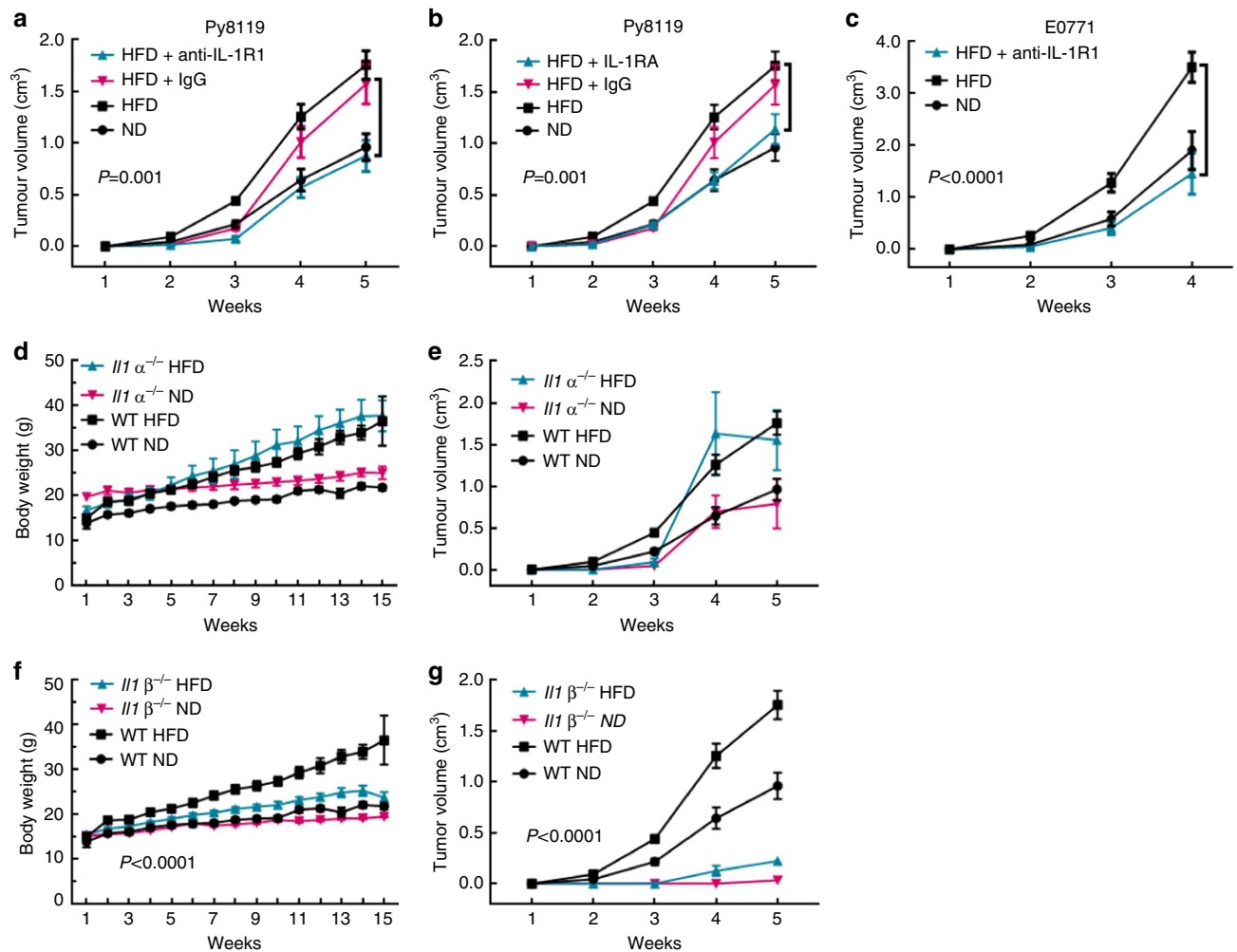

**Figure 1 | The IL-1β/IL-1R1 axis promotes tumour growth in diet-induced obese mice.** The indicated mice were given either a ND or HFD for 10 weeks, and then the indicated cells were implanted into the #4 mammary gland. See Supplementary Fig. 2b. (**a–c**) Mice were treated with anti-IL-1R1 antibody or control IgG once tumours were palpable (**a,c**) or with recombinant IL-1 receptor antagonist (rIL1RA) (**b**) starting the day of Py8119 cell transplant. Data represent the average tumour volume ± s.e.m ($a$–$b$: $n = 5$ HFD + IgG, $n = 5$ HFD + anti-IL-1R1, $n = 5$ HFD + anti-IL1RA $n = 14$ ND, $n = 24$ HFD; C: $n = 5$ all groups). (**d,e**) Data represent the average body weight (**d**) and Py8119 tumour volume (**e**) of the indicated mice ± s.e.m. ($n = 5$ $Il1\alpha^{-/-}$ ND, $n = 4$ $Il1\alpha^{-/-}$ HFD, $n = 14$ WT ND, $n = 24$ WT HFD). (**f,g**) Data represent the average body weight (**f**) and tumour volume (**g**) of the indicated mice ± s.e.m. (($e$) $n = 5$ $Il1\beta^{-/-}$ ND, $n = 5$ $Il1\beta^{-/-}$ HFD, $n = 14$ WT ND, $n = 24$ WT HFD; f: $n = 1$ $Il1\beta^{-/-}$ ND, $n = 2$ $Il1\beta^{-/-}$ HFD). Only mice that had measurable tumours after 5 weeks are represented in **g**. Two-way ANOVA was used to determine significance. All tumour studies were repeated in a different cohort of animals.

under obesity. We first examined the role of CASP1 in ODBP. Consistent with previous reports[10], $Casp1/11^{-/-}$ mice ($Casp1^{-/-}$ mice also have a concomitant loss of $Casp11$, referred to as $Casp1/11^{-/-}$) had increased weight gain with HFD, similar to that seen in WT mice (Fig. 2a). $Casp1/11^{-/-}$ DIO mice had significantly reduced tumour growth compared with WT DIO mice (Fig. 2b), but CASP1/11 deficiency had no impact on tumour growth in normal-weight mice (Fig. 2b), suggesting that simply gaining body weight in the $Casp1/11^{-/-}$ mice is no longer sufficient for tumour growth. These data suggest that similar to the IL-1 blockade, only the obesity-associated tumour growth depends on CASP1. To examine whether CASP1 modulates ODBP in NLRP3 inflammasome-dependent manner, we obtained $Nlrp3^{-/-}$ mice and found that, while the mice lacking NLRP3 had less weight gain with HFD than WT mice (Supplementary Fig. 3), NLRP3 deficiency had no significant impact on tumour growth in DIO or normal-weight mice (Fig. 2c). The NLRC4 inflammasome has been shown to be activated only by bacterial products[20]; as such, $Nlrc4^{-/-}$ mice were initially included as control. To our

surprise, $Nlrc4^{-/-}$ DIO mice exhibited a decrease in tumour growth similar to the $Casp1/11^{-/-}$ DIO mice, relative to WT DIO counterparts (Fig. 2d,e, $P = 0.005$, two-way analysis of variance (ANOVA)). $Nlrc4^{-/-}$ DIO mice had significantly increased body weight (Supplementary Fig. 3). There was no significant difference in Py8119 tumour growth between normal-weight $Nlrc4^{-/-}$ and WT mice (Fig. 2d). Similarly, E0771 tumour growth in $Nlrc4^{-/-}$ DIO mice was significantly reduced compared with their WT DIO counterparts (Fig. 2f, $P = 0.001$, two-way ANOVA). In contrast to Py8119 tumours, E0771 tumour growth was also reduced in $Nlrc4^{-/-}$ relative to WT mice with ND (Fig. 2f), suggesting a slightly different phenotype and indicating that the NLRC4 inflammasome could be also important for tumour growth in non-obese condition for certain cancers.

**Obesity induces NLRC4 inflammasome in myeloid cells.** Inflammasomes are normally activated from innate immune cells. We thus examined tumour-infiltrating immune cells in DIO

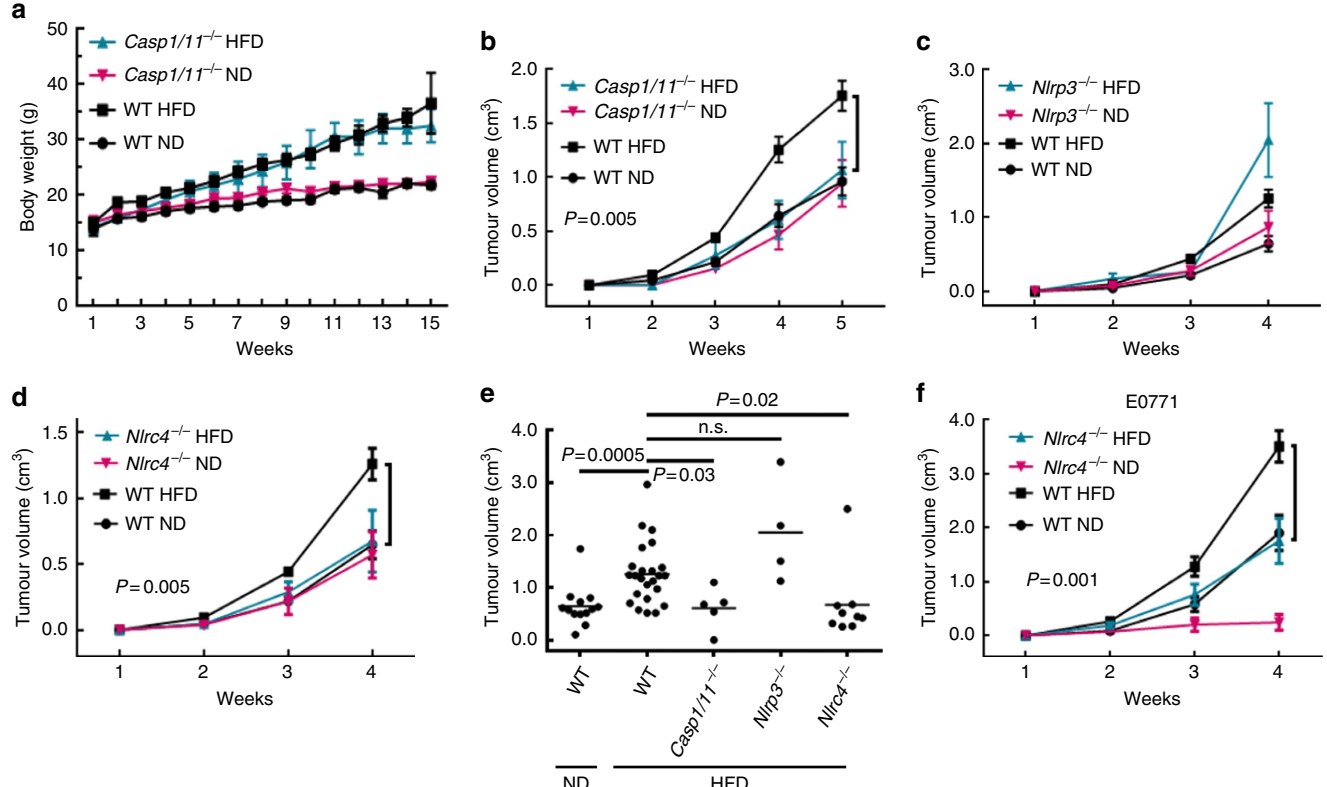

**Figure 2 | NLRC4 inflammasome promotes tumour growth in diet-induced obese mice.** (**a–d**) The indicated mice were treated as in Fig. 1. Data represent body weight (**a**) in $Casp1/11^{-/-}$ mice versus WT mice) or the average Py8119 tumour volume ± s.e.m. (**b–d**; $n = 5$ $Casp1/11^{-/-}$ ND, $n = 5$ $Casp1/11^{-/-}$ HFD, $n = 5$ $Nlrp3^{-/-}$ ND, $n = 4$ $Nlrp3^{-/-}$ HFD, $n = 5$ $Nlrc4^{-/-}$ ND, $n = 9$ $Nlrc4^{-/--}$ HFD $n = 14$ WT ND, $n = 24$ WT HFD). Two-way ANOVA was used to determine significance. (**e**) Py8119 tumour volumes 4 weeks after implantation from the indicated mice from **a** to **d**. Indicated $P$ values are from a Student's $t$-test comparing individual group means. One-way ANOVA was performed across all groups: $P < 0.0001$. (**f**) Average E0771 tumour volume ± s.e.m. from the indicated mice ($n = 5$ WT ND, $n = 5$ WT HFD, $n = 5$ $Nlrc4^{-/-}$ ND, $n = 9$ $Nlrc4^{-/-}$ HFD). Two-way ANOVA was used to determine significance. All tumour studies were repeated in a different cohort of animals.

tumours compared with ND tumours using flow cytometry. We found no difference in $CD4^+$ and $CD8^+$ T cells, $B220^+$ B cells, monocytic (Ly-6C$^+$ CD11b$^+$) or granulocytic (Ly-6G$^+$CD11b$^+$) myeloid-derived suppressor cells/neutrophils. We did observe a significant increase in the number of tumour-infiltrating macrophages (F4/80$^+$CD11b$^+$); in particular, there was a significant increase in CD11c$^-$ macrophages and a trend for increased CD11c$^+$ macrophages[21] (Fig. 3a). We further found an increase in regulatory cytokines including $Il10$ and $Arg1$, but not pro-inflammatory cytokines such as $Il6, Tnfa$ and $Il12$ in DIO tumours compared with normal tumours using purified CD11b$^+$ myeloid cells (Supplementary Fig. 4a,b). We observed a statistically significant increase in $Il1b$, but not $Il1a$ expression in Py8119 tumours from DIO mice compared with ND mice, but no difference in the expression of other inflammatory cytokines (Fig. 3b and Supplementary Fig. 4c). Similar results were obtained for E0771 tumours where we observed a significant increase in the expression of $Il1b$ (Supplementary Fig. 4d). We did observe a discrepancy in the $Il18$ mRNA level from the two models (Supplementary Fig. 4c,d), further excluding the involvement of IL-18, the other inflammasome substrate, in ODBP. Inflammasome activation is indicated by the processing of CASP1 into the active p10 and p20 subunits[22]. We found that tumour-infiltrating CD11b$^+$ myeloid cells from DIO mice had increased CASP1 processing but neither in CD11b$^+$ cells from ND tumours nor in CD11b$^-$ cells (Fig. 3c and Supplementary Fig. 4e, left panels). As our $Casp1^{-/-}$ mice lack $Casp11$ (ref. 23), we also examined CASP11 processing in

CD11b$^+$ and CD11b$^-$ cell populations. We found that CASP11 was primarily present in CD11b$^+$ cells; however, the cleaved CASP11 was lower in tumours from HFD mice (Fig. 3c and Supplementary Fig. 4e, right panels), suggesting that CASP1 but not CASP11 is activated in tumour-infiltrating myeloid cells under obesity.

We identified 4.1-fold increase in $Nlrc4$ mRNA but not $Nlrp3$ expression in tumours from DIO mice compared with those from ND mice (Fig. 3d). There was very little $Nlrc4$ mRNA in tumours from $Nlrc4^{-/-}$ DIO mice (Fig. 3d), indicating that $Nlrc4$ is primarily expressed in the host but not in the cancer cells. We found that tumour-infiltrating CD11b$^+$ cells were the primary source of $Nlrc4$ in both Py8119 (Fig. 3e) and E0771 (Fig. 3f) tumours, and that obesity further upregulated its expression (Fig. 3e,f, open bars). Using NLRC4-Flag knock-in mice[24] we detected two isoforms of NLRC4, isoform 1 (114 kd) and isoform 2 (44 kd) in tumour-infiltrating CD11b$^+$ cells from DIO mice, but no NLRC4 expression could be detected in CD11b$^-$ cells (Fig. 3g and Supplementary Fig. 4f). Using a fluorescent probe for active CASP1, we also found that obesity-induced CASP1 activation was significantly reduced in CD45$^+$ tumour-infiltrating leukocytes from NLRC4-deficient DIO mice compared with WT DIO mice (Fig. 3h). To further confirm the role of macrophage-specific NLRC4 inflammasome activation in ODBP, we co-injected WT or $Nlrc4^{-/-}$ bone marrow-derived macrophages (BMDMs) with Py8119 cells orthotopically into $Nlrc4^{-/-}$ DIO mice. Co-injection of cancer cells with WT BMDM had significantly increased tumour growth relative

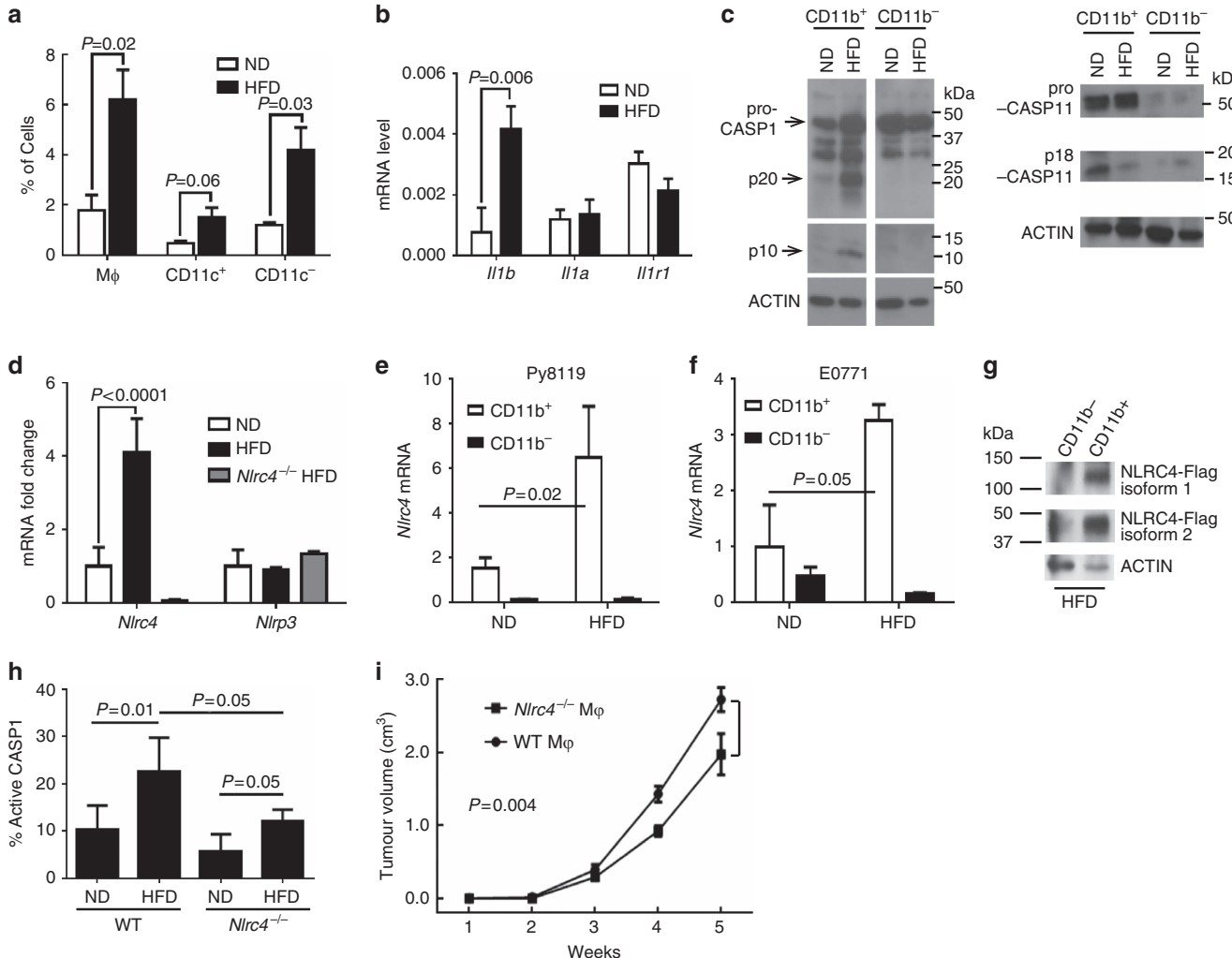

**Figure 3 | Obesity-induced NLRC4 inflammasome in tumour-infiltrating myeloid cells. (a)** Data represent the number of tumour-infiltrating total (mφ), CD11c$^-$ or CD11c$^+$ macrophages as a percentage of total cells counted ± s.d. ($n = 5$ ND; $n = 9$ HFD). **(b)** mRNA expression (relative to *peptidylprolyl isomerase A* gene; *Ppia*) of the indicated genes in tumours from indicated mice ($n = 3$) ± s.d. **(c)** Western blot analysis for CASP1 (left panels) and CASP11 (right panels) in tumour-infiltrating CD11b$^+$ and CD11b$^-$ cell populations. Cells were combined from four to five tumours from each group. **(d)** mRNA fold-change relative to the ND group, using *Ppia* as reference gene. Mean ± s.d. in tumours from the indicated mice ($n = 3$). **(e,f)** *Nlrc4* mRNA expression in tumour-infiltrating CD11b$^+$ and CD11b$^-$ cell populations from the indicated mice in the indicated tumour model, using *Actin beta* (*Actb*) as the reference gene. Cells were combined from four to five tumours from each group, and data are shown in triplicates. **(g)** Western blot for NLRC4-flag in tumour-infiltrating CD11b$^-$ and CD11b$^+$ cell populations from DIO mice. **(h)** Data represent the average number of tumour-infiltrating CD45$^+$ cells with CASP1 activation as a percentage of CD45$^+$ cells ± s.d. ($n = 5$ WT ND, $n = 5$ WT HFD, $n = 5$ *Nlrc4$^{-/-}$* ND, $n = 4$ *Nlrc4$^{-/-}$* HFD). For all panels, Student's *t*-test was used to determine significance. **(i)** NLRC4 activation from macrophages drives ODBP. Bone marrow macrophages from WT or *Nlrc4$^{-/-}$* female mice were co-injected with Py8119 cells orthotopically into DIO *Nlrc4$^{-/-}$* female mice. Tumour growth was monitored weekly ($n = 7$, means ± s.e.m.). Tumour study was repeated in a different cohort of animals. All other studies represent results from two to three repeats.

to mice co-injected with *Nlrc4$^{-/-}$* BMDM (Fig. 3i). These data support the sufficiency of myeloid NLRC4 inflammasome activation in driving disease progression under obesity.

**NLRC4/IL-1β module promotes angiogenesis in DIO mice.** To determine whether the IL-1β module acts directly on the tumour cells, we silenced *Il1r1* in Py8119 cells and found no impairment of tumour growth in DIO mice (Supplementary Fig. 5a), indicating that IL-1β does not primarily act on cancer cells. We examined a panel of angiogenesis-associated proteins and identified a marked increase in many angiogenesis-associated proteins in tumour lysates from DIO mice relative to those from ND mice (Fig. 4a, left panels, red boxes indicate reference spots for loading), including the pro-angiogenic inflammatory

chemokines CXCL1, CCL2 and CCL3 (refs 14,25,26; Fig. 4a, right panel). Other proteins that were increased in tumours from DIO mice included MMP9 that is known to promote cancer progression[27] and NOV (CCN3) that has been associated with obesity-related metabolic disorders[28] (Fig. 4a, right panel). We also observed a significant increase in CD31$^+$ endothelial cells in DIO tumours (Fig. 4b,c, white arrows for brown CD31 staining) in an NLRC4 inflammasome-dependent manner as blockade of IL-1R1 or the loss of the NLRC4 inflammasome (*Casp1/11$^{-/-}$* or *Nlrc4$^{-/-}$*) reduced CD31$^+$ staining in Py8119 tumours from DIO mice (Fig. 4b,c). These data were confirmed by immunofluorescent staining for CD31 (Supplementary Fig. 5b,c), as well as in the E0771 tumours where obesity increased CD31$^+$ staining in E0771 tumours from DIO mice in an IL-1/IL-1R1-dependent manner (Fig. 4d). We detected a 2.6-fold increase in

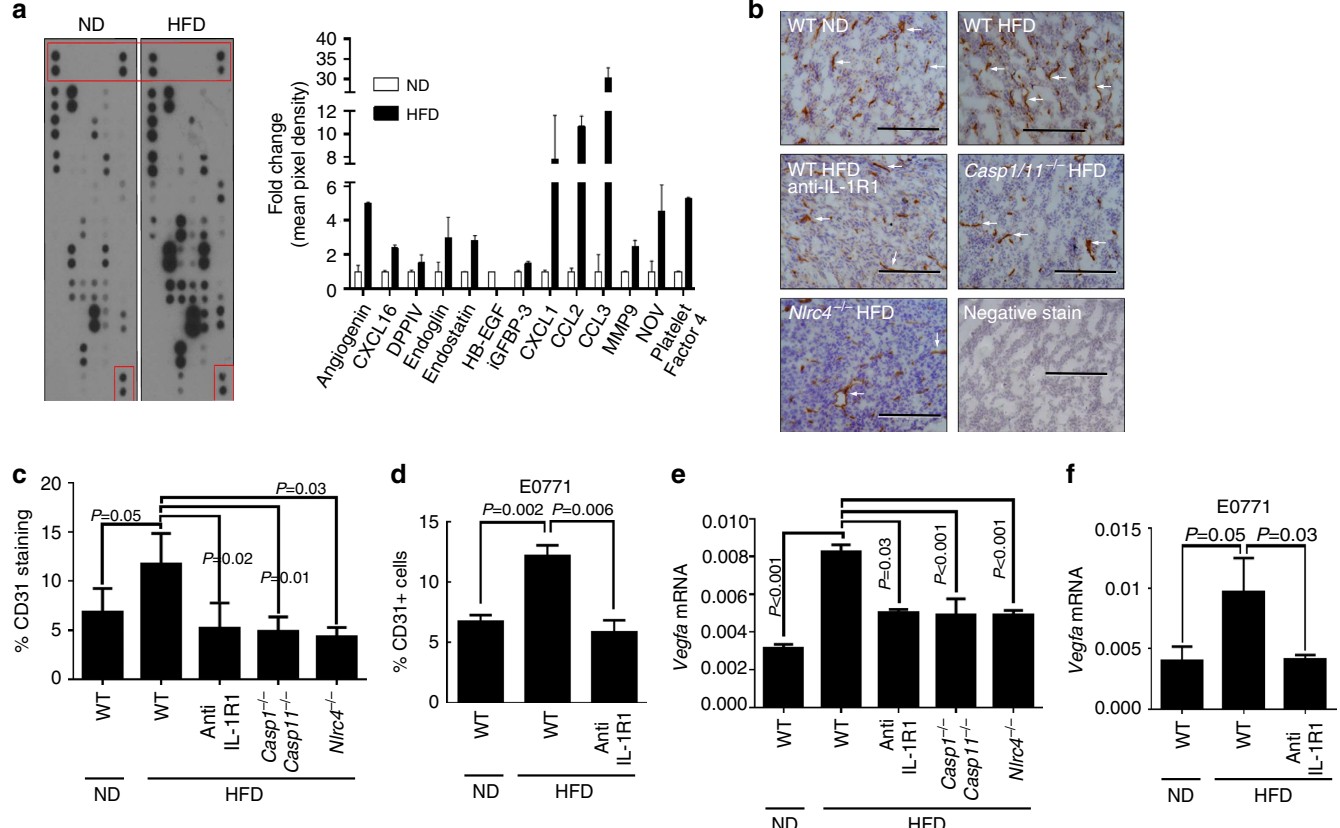

**Figure 4 | NLRC4 inflammasome promotes angiogenesis in diet-induced obese mice.** (**a**) Left panel: proteome profiler array for angiogenesis proteins using tumour lysates from ND and HFD mice (red boxes indicate the internal reference spots). Right panel: analysis of proteome profiler. Data represent the fold change compared with ND of the average mean pixel density ± s.d. of select proteins. (**b**) Representative immunohistochemistry (IHC) staining for CD31 (brown colour, some indicated with white arrows) from the indicated Py8119 tumours, haematoxylin (blue) being used for background nuclear staining. An isotype-negative control staining is included (negative control). Scale bar, 200 μm. (**c**) Quantification of IHC staining in **b**. Data represent the average area of CD31-positive staining over total area ± s.d. At least three fields per section and three tumours per group were used in the analysis. (**d**) Data represent the average number of CD31⁺ cells as a percentage of total cells counted by flow cytometry ± s.d. in E0771 from tumours of the indicated mice (n = 5 all groups). (**e,f**) Average *Vegfa* mRNA expression relative to *Ppia* in Py8119 tumours (**e**) and E0771 tumours (**f**) from the indicated mice ± s.d. (n = 3 for each group). All studies represent results from two to three repeats. For all panels, the group means were compared by Student's *t*-test to determine significance.

*Vegfa* mRNA in tumours from DIO mice compared with those from ND mice (Fig. 4e), in agreement with previous publications showing the positive correlation between BMI and serum VEGFA protein levels in human[29,30] and mouse[30,31]. This *Vegfa* upregulation was dependent on the NLRC4 inflammasome as it was abrogated in tumours from DIO mice where IL-1R signalling was blocked or the NLRC4 inflammasome was missing (Fig. 4e). Consistent with these findings, a similar increase in *Vegfa* mRNA was found in E0711 tumours from DIO mice compared with those from ND mice, which was also reduced by treatment with anti-IL-1R1 antibody (Fig. 4f). We also measured the expression of other *Vegf* family members in Py8119 tumours and did not find any statistically significant differences in the expression *Vegfb* or *Vegfc* (Supplementary Fig. 5d). We examined the expression of the chemokines whose levels were elevated in our proteome profiler array (Fig. 4a) including CXCL1, CCL2 and CCL3. We found that *Cxcl1* mRNA was elevated in tumours from obese mice but not in the NLRC4 inflammasome-dependent manner (Supplementary Fig. 5e). *Ccl2* and *Ccl3* mRNA levels were not increased in tumours from HFD mice (Supplementary Fig. 5e).

**IL-1β promotes *Vegfa* expression in adipocytes.** We further investigated the cellular source that mediates the pro-angiogenic

effect of IL-1β. We found that *Vegfa* expression was significantly increased in the CD11b⁻ population in tumours from DIO mice (Fig. 5a). We assessed the ability of IL-1β to induce the expression of *Vegfa in* different cell types, including cancer-associated fibroblasts, endothelial cells, primary mammary adipocytes and primary BMDM. Among all these cell types, we found that IL-1β induced profound elevation of *Vegfa* expression in primary adipocytes (10.4-fold increase, Fig. 5b), and to a lesser extent in SVEC endothelial cells (Fig. 5c), but not in WT primary BDMD (Mφ), or two cancer-associated fibroblasts (CAF1 and CAF2) established from our previous study[32] (Fig. 5c). It has been shown that macrophages can secrete angiogenic factors other than IL-1β[33]. To determine whether IL-1β could indirectly have an impact on angiogenesis through macrophages, we collected conditioned medium (CM) from non-treated or IL-1β-treated BMDM. We found that IL-1β-treated CM from macrophages induced a more robust elevation of *Vegfa* mRNA from adipocytes (35-fold increase compared with non-treated, sevenfold higher than that induced by the non-treated CM; Fig. 5b). These data indicate that IL-1β promotes *Vegfa* expression in adipocytes directly as well as indirectly through effects on macrophages. We further found that IL-1β-induced JNK activation is critical to promote *Vegfa*, whereas the IKK/NFκB pathway has very minor effect[34] (Fig. 5d). In support of this conclusion, adipocytes treated

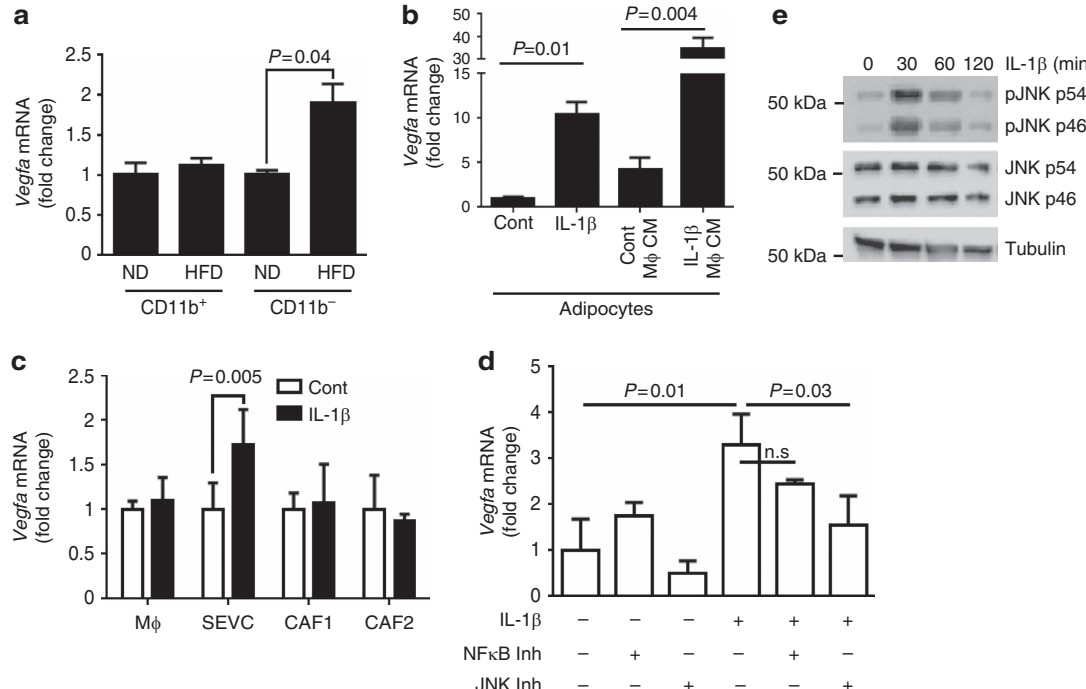

**Figure 5 | IL-1β-induced *Vegfa* expression in adipocytes.** (**a**) *Vegfa* expression is increased in non-myeloid cells under obesity. CD11b$^+$ and CD11b$^-$ cells were purified by magnetic beads in tumours from ND or DIO mice. Data represent the mean fold change compared with ND ± s.d. Expression is relative to *Actb* (n = 3 for each group). (**b**,**c**) Indicated cells were left untreated (cont), treated with 100 ng ml$^{-1}$ rIL-1β or CM from non-treated or 100 ng ml$^{-1}$ rIL-1β-treated primary BMDM. Data represent the average mRNA expression of *Vegfa* relative to *Ppia* as fold change compared with cont ± s.d. (**d**) Adipocytes were treated as indicated with 100 ng ml$^{-1}$ rIL-1β, 5 μM NFκB inhibitor (BMS345541) and 40 μM JNK inhibitor. Data represent the average mRNA expression of *Vegfa* relative to *Ppia* ± s.d. (n = 3 for all groups). (**e**) Primary mammary adipocytes were treated with 100 ng ml$^{-1}$ rIL-1β for the indicated time, and the indicated proteins were separated by SDS–PAGE followed by immunoblotting with the indicated antibodies. All studies represent results from two to three repeats. For all panels, the group means were compared by Student's *t*-test to determine significance.

with IL-1β had an increased JNK activation as indicated by their phosphorylation (Fig. 5e and Supplementary Fig. 5f). As these data suggest that NLRC4 inflammasome/IL-1β drives obesity-associated tumour angiogenesis through a signalling network between macrophages and adipocytes, we stained tumour sections for adipocytes and found a general increase in lipid droplets of tumours from DIO mice compared with ND mice (Supplementary Fig. 5g).

**Metformin inhibits obesity-induced angiogenesis in cancer.** Metformin, a type 2 diabetes drug, reduces the risk of several types of cancer including breast cancer[35–37]. Metformin is known to have pleiotropic effects, and is thought to primarily work by inhibiting mitochondrial respiratory-chain complex-I, which leads to activation of adenosine monophosphate-activated protein kinase[38]. We found that metformin specifically inhibited tumour growth in DIO mice (Fig. 6a) at the same time preventing weight gain in mice fed with HFD (Fig. 6b). In contrast, metformin had no effect on tumour growth or body weight of mice with ND (Fig. 6a,b). Metformin had no direct impact on NLRC4-inflammasome activity or the expression of *Il1b* (Supplementary Fig. 6a,b); however, it significantly reduced CD31$^+$ endothelial cells (Fig. 6c,d) as well as the expression of *Vegfa* (Fig. 6e). This is consistent with previous studies that have shown that metformin lowers serum VEGFA levels in both human and mouse[39,40]. Metformin had no effect on the expression of *Cxcl1* in tumours from HFD mice (Supplementary Fig. 5e). We also found that metformin inhibits the IL-1β-induced *Vegfa* expression in adipocytes (Fig. 6f), suggesting that

metformin may inhibit obesity-induced angiogenesis by inhibiting IL-1β-induced upregulation of *Vegfa*. These data suggest that angiogenesis is the common target for NLRC4/IL-β deficiency and metformin treatment.

**NLRC4 inflammasome in obesity and human breast cancer.** We next evaluated the relevance of the NLRC4 inflammasome and angiogenesis to breast cancer and obesity in human patients. Analysis of Geo Data set GSE33256 showed a significant increase in the expression of *NLRC4* in the obese breast tissues (Fig. 7a) and a positive correlation between *NLRC4* and BMI in normal human breast tissues (Fig. 7b). Analysis of the Cancer Genome Atlas (TCGA) data set showed that *NLRC4* is maintained or increased across the PAM50 subtypes compared with normal human breast tissues (Fig. 7c, left panel), while *NLRP3* expression is generally decreased in all subtypes (Fig. 7c, right panel). In addition, we found a significant increase in the expression of *CASP1* in basal and luminal B type tumours in obese patients (Fig. 7d). *CASP1* expression was also upregulated in these types of tumours from overweight patients (Fig. 7d). We did not see any correlation between BMI and *CASP1* expression in HER2 or Luminal A type breast cancer (Fig. 7d). The KMPlot analysis[41] indicated that *NLRC4* mRNA was inversely associated with overall survival in all breast cancer patients and even more so in ER+ luminal breast cancers (Fig. 7e, left two panels), indicating that *NLRC4* is a marker for poor prognosis. In contrast, *NLRP3* had a positive correlation with overall survival in breast cancer patients (Fig. 7e, right two panels). In addition, we examined TCGA invasive breast cancer expression data and also found that

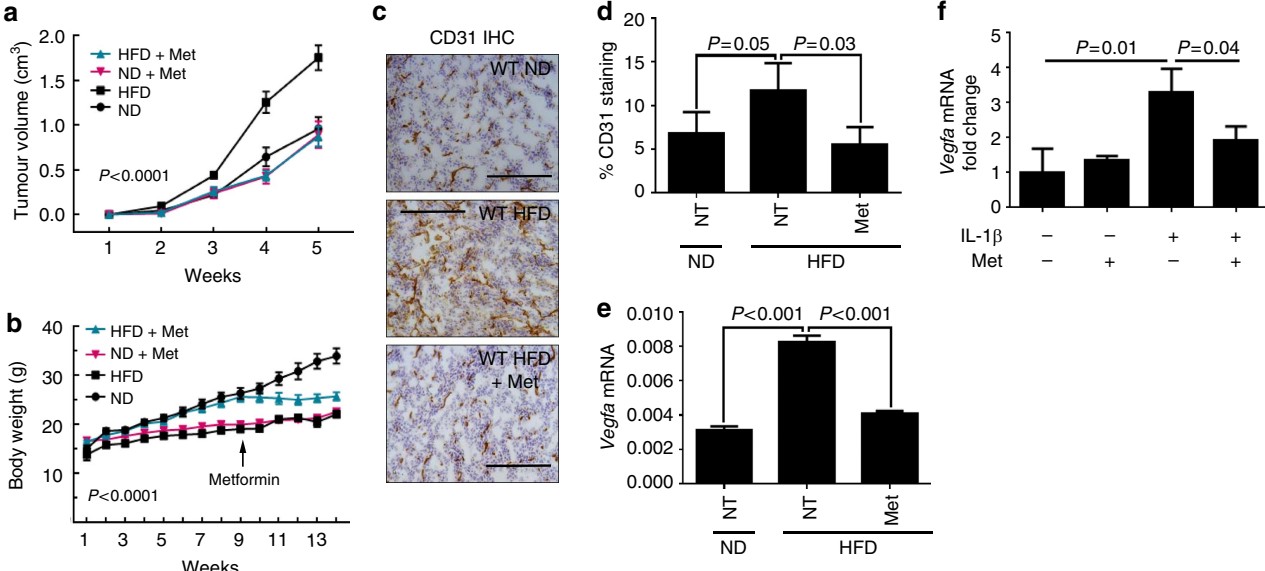

**Figure 6 | Metformin inhibits tumour growth and angiogenesis in diet-induced obese mice.** Mice were treated as in Fig. 1a; except after 9 weeks one group of ND and one group of HFD mice were fed with 0.5% metformin water. (**a,b**) Average Py8119 tumour volume (**a**) or body weight (**b**) ± s.e.m. ($n = 5$ ND + Met, $n = 10$ HFD + Met, $n = 14$ ND, $n = 24$ HFD). Two-way ANOVA was used to determine significance. (**c**) Representative IHC staining for CD31 from the indicated mice. Scale bar, 200 μm. (**d**) Quantification of IHC staining in **c**. Data represent the average area of CD31+ staining over total area ± s.d. At least three fields per section and three tumours per group were used in the analysis. (**e**) Average mRNA expression of *Vegfa* ± s.d. in tumours from the indicated mice ($n = 3$). (**f**) Adipocytes were treated with 100 ng ml$^{-1}$ rIL-1β and 500 μM metformin as indicated. Data represent the average *Vegfa* mRNA level relative to *Ppia* as a fold change compared with non-treated cells ± s.d. ($n = 3$). Tumour study was repeated in a different cohort of animals. All other studies represent results from two to three repeats. Group means were compared by Student's *t*-test to determine significance.

highest *NLRC4* tertile was significantly associated with shorter overall survival (Supplementary Fig. 7a), while *NLRP3* expression had no correlation with survival (Supplementary Fig. 7b). We further confirmed a strong positive correlation between *NLRC4* expression with myeloid lineage markers from the TCGA data set, including specific macrophage marker CD163 and common myeloid markers CD68 and CD33, suggestive of a myeloid origin of NLRC4 in human cancer (Fig. 7f). We also found a positive correlation between *NLRC4* and its target *IL1β* (Fig. 7f). These data support the findings in our mouse models that the NLRC4 inflammasome in myeloid cells promotes obesity-associated disease progression.

## Discussion

There have been several hypotheses to explain the link between obesity and breast cancer development and progression; however, most of these studies are correlative, if not all, and have not established the causal mechanism because of the lack of using genetically modified mouse models with an intact immune system. Here we identified a causal link between obesity and breast cancer progression. Obese tumour microenvironment recruits macrophages (likely via CCL2 chemotaxis[14,42]) with an activated NLRC4 inflammasome that leads to IL-1β activation. IL-1β then promotes disease progression at least in part by acting on adipocytes directly, or induces secreted factors from macrophages indirectly to promote adipocyte-originated VEGFA production and angiogenesis. Furthermore, we show that treatment of obese mice with metformin inhibits obesity-associated tumour growth via modification of the tumour microenvironment through the reduction in obesity-associated angiogenesis (Fig. 6). The inhibition of obesity-induced tumour growth and angiogenesis by metformin precisely phenocopied the

blockade of IL-1 signalling, suggesting that metformin may act within this pathway, possibly by blocking IL-1β-mediated upregulation of *Vegfa* in adipocytes (Fig. 6f). This is supported by other studies that have shown that metformin can inhibit expression of IL-1β- and IL-1-dependent genes[43,44]. However, at this time the *Nlrc4*-activating signal remains unclear as the only previously identified activators of NLRC4 are bacterial flagellin and components of bacterial secretion systems[45]. HFD and genetic loss of *NLRC4* can alter the gut microbiota, and HFD can lead to increased endotoxaemia, all of which contributes to obesity-associated inflammation[46,47]. Therefore, it is possible that these changes in the gut microbiota and endotoxaemia promote NLRC4 inflammasome activation. Even though we mixed dirty beddings from each cage and added to clean cages to minimize the potential microbiota change, the strict co-housing experiments were prohibited by feeding different cages with ND or HFD. Future studies will be needed to determine whether NLRC4 is activated because of changes in the gut microbiota, endotoxaemia or some tumour-associated protein that act as a homologue to these NLRC4-activating bacterial components. Our findings provide a rationale to develop therapeutics specifically for treating obese cancer patients. While anti-VEGF therapy Avastin is not effective in breast cancer and the use of metformin in breast cancer is still undergoing in several clinical trials focused on obese patients[48], these drugs still hold promise to be used in selective obese patient populations. Most importantly, drugs targeting IL-1β signalling such as anakinra (Fig. 1) or canakinumab (anti-IL-1β) are approved for treating other diseases and can be adapted to obese cancer patients. Other inhibitors, including long-lasting CASP1 inhibitors[49] and other angiogenesis inhibitors, may also be beneficial for treating obese cancer patients with a high basal level of neoangiogenesis as has been recently indicated[50], along with current standard therapies.

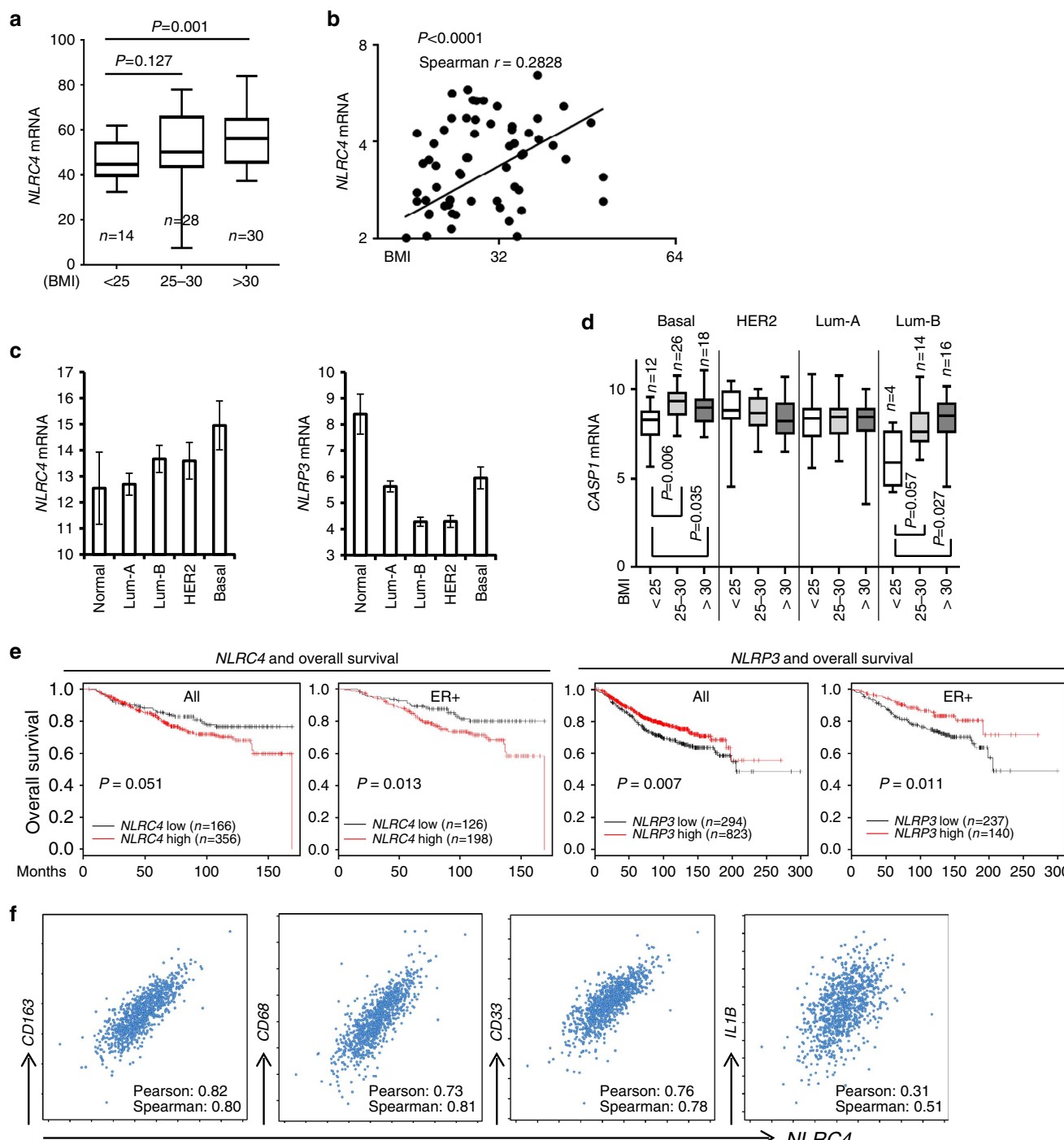

**Figure 7 | NLRC4 and CASP1 are associated with macrophage markers and poor outcome in human breast cancer.** (**a**,**b**) *NLRC4* mRNA is elevated in normal breast tissues from obese individuals (**a**) and is positively correlated with BMI (**b**). Number of cases indicated. Data from GSE33256 GEO data set. Statistical significance was determined by Welch's *t*-test (**a**) and linear regression and F test (**b**). (**c**) The mean *NLRC4* and *NLRP3* expression ± 95% confidence interval (CI) in PAM50 subtypes of breast and normal breast tissue from TCGA-invasive breast cancer data set. Number of cases indicated. (**d**) *CASP1* mRNA is elevated in obese breast cancer patients with luminal B and basal-like breast cancers. Data are presented as the mean ± 95% CI. Data are from GSE20194 GEO data set. Statistical significance was determined by Welch's *t*-test. (**e**) Correlation of *NLRC4* expression (left two panels) or NLRP3 expression (right two panels) with overall survival in months within all human breast cancer patients (all) or within the ER + breast cancers (ER + ), analysed using KM-Plot meta-data set for invasive breast cancer. Statistical significance was determined by log-rank test comparing low versus high groups with number of cases indicated. (**f**) *NLRC4* expression is positively correlated with macrophage markers including *CD163* and common markers for myeloid cells such as *CD68* and *CD33*, and is also correlated with *IL1B* expression in human breast cancer from TCGA (n = 960 analysed from cBioPortal). Spearman and Pearson *r* values are indicated.

## Methods

**Cell lines and cell culture.** Py8119 cells were derived from a primary *MMTV-PyMT* tumour in the C57BL/6 background as described previously[16]. E0771 cells, provided by Dr Mikhail Kolonin (UT Health Science Center at Houston), were isolated from a spontaneous breast adenocarcinoma in C57BL/6 mice[17]. Cancer-associated fibroblast cell lines (CAF1 and CAF2) were isolated from primary tumours from *MMTV-ErbB2* mice and described previously[32]. SVEC cells[51] were purchased from American Type Culture Collection. Py8119 cells were

maintained in F12 media supplemented with 10% fetal calf serum (FCS), 10 ng ml$^{-1}$ epithelial growth factor, 2 µg ml$^{-1}$ hydrocortisone and 5 µg ml$^{-1}$ insulin. CAF1, CAF2, E0771 and SVEC cells were maintained in Dulbecco's modified Eagle medium supplemented with 10% FCS. For experiments involving IL-1β treatment, cells were treated with 100 ng ml$^{-1}$ recombinant mouse IL-1β (eBiosciences, San Diego, CA, USA) for 6 h. E0771 and Py8119 cells were tested negative for mycoplasma using the Mycoalert Mycoplasma Detection Kit (Lonza).

**Mouse colony and orthotopic transplant model.** All animal experiments were performed on protocols approved by the University of Iowa Institutional Animal Care and Use Committee (IACUC) and in accordance with the University of Iowa IACUC guidelines. All mice were females of a C57BL/6N background. Wild-type C57BL/6N mice were purchased from Charles River (Wilmington, MA). Il1α$^{-/-}$, Il1β$^{-/-}$, Nlrc4$^{-/-}$, Nlrp3$^{-/-}$ and Casp1$^{-/-}$ mice were provided by Dr Fayyaz Sutterwala[23,52–55]. C57BL/6N OVX mice were purchased from NCI. NLRC4-Flag knock-in mice were obtained from Genentech (San Francisco, CA)[24].

For the orthotopic transplant model, 6-week-old mice were fed a HFD (BioServ, Flemington, NJ), an ND (Harlan) provided by the vivarium or a nutrient Cont ND (from BioServ, composed of the same macronutrients as the HFD, except the HFD diet is 36% fat and 35.7% carbohydrates, while the Cont ND is 7.2% fat and 61.6% carbohydrates) and body weight was monitored weekly. A minimum of four mice per group were used in all experiments in order to obtain statistical significance, and all tumour studies were repeated with a different cohorts of animals. Mice for each experiment were randomly assigned to groups receiving ND, Cont ND or HFD. After 10 weeks, $1 \times 10^5$ Py8119 or $2 \times 10^5$ E0771 cells resuspended in 50/50 matrigel/PBS were transplanted into the #4 mammary fatpad. Tumour volume was monitored weekly and calculated as length $\times$ width$^2$ $\times$ 0.5. Mice were killed and tumours were collected after 4 or 5 weeks. The mice in Fig. 1a were treated with 2 mg kg$^{-1}$ anti-IL-1R1 antibody (BioXcell, West Lebanon, NH) or Hamster IgG (BioXcell) by i.p. injection once tumours were palpable (2–3 mm in length). Mice were then treated 5 days later, following with once a week thereafter, with the same dose. Mice in Fig. 1b were treated with 5 mg kg$^{-1}$ recombinant interleukin-1 receptor antagonist (Swedish Orphan Biovitrim, Stockholm, Sweden) by daily i.p. injection starting on the same day when the cells were injected. The mice treated with metformin (Fig. 6) were given 0.5% metformin in their drinking water 1 week before cells were transplanted. Owing to the nature of different genetics and treatment, it is impossible for double-blinded experiments for animal research. Most of the analyses, however, were confirmed at least by a second person for the accuracy of the measurements.

**Single-cell isolation and flow cytometry.** Single cells from tumours were isolated as described previously[32]. Briefly, tumours were digested with 300 µg ml$^{-1}$ collagenase and 100 µg ml$^{-1}$ hyaluronidase (Stemcell Vancouver, BC, Canada), 0.25% trypsin (Mediatech, Corning, NY) and 0.1 mg ml$^{-1}$ DNase I (Worthington, Lakewood, NJ), filtered through 40-µm mesh and resuspended in HBSS with 2% FCS. Single cells were then stained with antibodies including the following: anti-mouse CD11c-Pac Blue (N418; 2.5 µg ml; Biolegend, San Diego, CA), anti-mouse F4/80-PE (BM8; 2 µg ml$^{-1}$), CD11b-FITC (M1/70; 2.5 µg ml$^{-1}$), CD45-PerCP (30-F11; 2 µg ml$^{-1}$), Ly6G-APC (1A8; 2.5 µg ml$^{-1}$) and Ly6C-PE-Cy7 (HK1.4; 2 µg ml$^{-1}$; Ebiosciences, CA). Cells were sorted with flow cytometry using an LSR II flow cytometer (BD Biosciences, Franklin Lakes, NJ) or an Accuri C6 flow cytometer (BD Biosciences). Analysis of flow cytometry data was done using FlowJo. CD11b$^+$ myeloid cells were purified using CD11b-positive selection kit (STEMCELL). Some other antibodies include the following: CASP1 p10 (M20, 1 µg ml$^{-1}$) antibody (Santa Cruz Biotechnology, Santa Cruz, CA); CASP1 antibody (14F468, 1 µg ml$^{-1}$, Genetex, Irvine, CA); M2 Flag antibody (Sigma-Aldrich, St Louis, MO, 0.5 µg ml$^{-1}$); Casp11 antibody (17D9, 1 µg ml$^{-1}$, Novus Biologicals, Littleton, CO); Phos-JNK (G-7, 1 µg ml$^{-1}$) and JNK (D-2, 1 µg ml$^{-1}$) from Cell Signaling Technologies (Danvers, MA).

**BMDMs and mammary adipocytes.** BMDMs were harvested and cultured from the indicated mouse strains. Briefly, bone marrow was flushed from the bones of the hind legs and cultured for 6–10 days in non-tissue-treated dishes in DMEM with 10% FCS plus conditioned media from L929 cells at a 1:5 ratio[53]. BMDM CM was collected after 72 h of 100 ng ml$^{-1}$ IL-1β treatment. Primary mouse pre-adipocytes were isolated from the mammary fat pad of 4-day-old mice and cultured in DMEM supplemented with 10% FCS, non-essential amino acids (Thermofisher, Waltham, MA), Glutamax (Thermofisher), 20 mM HEPES and 0.1 mM 2-mercaptoethanol. Preadipocytes were differentiated into mature adipocytes with 500 µM 3-isobutyl-1-methylxanine (Caymen Chemical Company, Ann Arbor, MI) and 1 µM Dethamexasone (Caymen Chemical Company) and maintained in DMEM supplemented with 10% FBS and 5 µg ml$^{-1}$ (ref. 56). Differentiated adipocytes were treated with 100 ng ml$^{-1}$ of recombinant mouse IL-1β (eBiosciences) for 6 h. For experiments involving co-treatment of adipocytes, cells were treated with 5 µM BMS345541 (Sigma-Aldrich), an NFκB inhibitor, 40 µM SP600125 (Sigma Aldrich) a JNK inhibitor or 500 µM metformin 1 h before treatment with recombinant mouse IL-1β.

**FAM/FLICA CASP1 activity assay.** Single cells were isolated from tumours and CASP1 activation was determined using FAM/FLICA CASP1 activity assay (Immunochemistry Technologies, Bloomington, MN) as per the manufacturer's instructions. Briefly, single cells were stained with FAM/FLICA-active CASP1 probe and CD45-PE (eBioscience). Cells were then sorted using flow cytometry for CD45$^+$/FLICA (GFP)$^+$ cells.

**Real-time PCR.** RNA from tumours and cells was isolated using the RNeasy Mini Plus Kit (Qiagen, Venlo, Limburg, the Netherlands) and reverse-transcribed using the iScript cDNA synthesis Kit (Bio-Rad, Hercules, CA). Real-time PCR was performed using iTaq Universal SYBR Green Supermix (Bio-Rad). Primers used are listed in Supplementary Table 1.

**Immunohistochemistry.** Tumours were collected and preserved in formalin or frozen in OCT compound. Paraffin sections were deparaffinized in xylene and antigens were retrieved with Antigen Unmasking Solution (Vector Laboratories, Burlingame, CA). Antibodies used for immunohistochemistry were anti-CD31 (MEC13.3, Biolegend 102502, 1 µg ml$^{-1}$). CD31 staining was quantified using the ImageJ software.

**Proteome profiler array.** Proteome profiler array was performed using the Mouse Angiogenesis Array Kit (R&D Systems) as per the manufacturer's instructions. Briefly, tissues were homogenized in PBS plus protease inhibitors and lysed with 1% triton X-100 and a freeze/thaw cycle. Equal amounts of protein lysates from three different tumours within each group were combined for experiments. Samples were incubated with provided membranes and proteins were visualized as described by the manufacturer. Proteins were quantified by measuring the mean pixel density of the individual spots and adjusted based on reference spots using the ImageJ software.

**Western blot analysis.** For western blot analysis, CD11b$^+$ and CD11b$^-$ cell populations were lysed with RIPA buffer (10 mM Tris, pH 8; 150 mM NaCl, 0.1% sodium deoxycholate; 1% Triton X-100, 1 mM EDTA; 1 mM phenylmethylsulphonyl fluoride, 1 µM dithiothreitol and protease inhibitors). Cell lysates were then separated by SDS–PAGE and analysed by standard western blotting protocol.

**Inflammasome activation ex vivo.** For inflammasome activation, $5 \times 10^5$ BMDMs were plated in 24-well plates and primed with 100 ng ml$^{-1}$ LPS-EK (Invivogen, San Diego, CA) for 4 h. Cells were then either treated with tumour homogenate in DMEM (Fig. 2h) for 24 h or transfected with 500 ng Flagellin-ST (Invivogen) for 6 h. For cells treated with metformin, the indicated dose of metformin was added 1 h before transfection with flagellin. The media was collected at the end of the experiment and IL-1β was measured by ELISA. Antibody pairs for the ELISA were from (R&D Systems).

**shRNA for IL-1R1.** cDNA for a short hairpin RNA (shRNA) for IL-1R1 were ligated into the lentiviral vector pLSLPw. Targeting sequence: 5′-CAAGTGTCCT CTTACTCCAAATAAA-3′. Virus particles were packaged and Py8119 cells were infected. Infected cells were selected for with puromycin. Lentivirus with no shRNA was used as a control. Knockdown of IL-1R1 was verified by SDS–PAGE and western blot.

**TCGA and microarray analyses.** TCGA breast cancer data set was downloaded and analysed as before[57] either via Cancer Genome Browser (https://genome-cancer.ucsc.edu/proj/site/hgHeatmap/) or via cBioPortal (http://www.cbioportal.org/). Expression analyses of the TCGA BRCA cohort were performed using the mean-centralized level 3 Illumina HiSeq2000 RNAseq data. Samples were separated into PAM50 subtypes for expression analyses. Kaplan–Meier curve for overall survival was generated using provisional overall survival data for all level 3 Illumina HiSeq expression data for primary tumours. Samples were split into tertiles based on Nlrc4 expression. GSE33256 and GSE20194 microarray data sets were obtained from NCBI GEO Data sets. Patient BMI data for GSE20194 were provided by Dr Sai-Ching Yeung (MD Anderson)[15]. Samples were separated based on PAM50 subtype and BMI. Microarray data for tumours from MMTV-TGFα;A$^y$/a and MMTV-TGFα;a/a mice were described previously[15]. For pathway analysis, data were analysed using Nexus Expression 3.0 (BioDiscovery) and Ingenuity Pathway Analysis (Ingenuity System). The signalling pathway graphs were generated using Ingenuity Pathway Analysis (Ingenuity System). In addition, KMPlot (http://kmplot.com/analysis/) was used to analyse microarray data sets that were normalized and combined for survival analysis[41].

**Statistics.** Data are presented as mean ± s.d. or 95% confident interval except otherwise indicated. Two-way ANOVA was used to compare tumour growth and mouse body weight as a function of time. Welch's t-test was used to determine significance in genomic data. One-way ANOVA with post hoc intergroup comparisons was used to determine significance for group difference with more

than two groups. For pathway analysis, a right-tailed Fisher's exact-test was used to determine significance. For all other data, individual group means were compared using a two-tailed Student's $t$-test to determine statistical significance. $P$ values of less than 0.05 were considered significant. Analyses were performed using the Prism software (GraphPad, San Diego CA).

**Data availability.** The TCGA data set referenced during the study is available in the public repository from the TCGA website (cancergenome.nih.gov). GSE33256 and GSE20194 microarray data sets referenced in the study were obtained from NCBI GEO Data sets (ncbi.nlm.nih.gov/gds). The data set from the $TGF\alpha;A^y/a$ and MMTV-$TGF\alpha;a/a$ mice referenced in the study was provided by Dr Sai-Ching Yeung (MD Anderson) and is available upon request. The authors declare that all the other data supporting the findings of this study are available within the article and its Supplementary Information files and from the corresponding author upon reasonable request.

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

## Acknowledgements

W.Z. was supported by NIH grants CA158055, CA200673 and CA203834, the V Scholar award, American Cancer Society seed grant, Breast Cancer Research Award and Oberley Award (National Cancer Institute Award P30CA086862) from Holden Comprehensive Cancer Center at the University of Iowa, and startup funds from the Department of Pathology, University of Iowa. F.S.S. was supported by NIH grant R56 AI118719. R.K. was supported by NIH T32 AI007260 and LGE by K22CA118182. A.M.J. was supported by NIH T32 AI007485. S.L.C. was supported by NIH R01 AI104706. S.-C.J.Y. and M.-H.L. were supported by the Susan G. Komen for the Cure Promise Grant (KG081048). We thank the Comparative Histopathology Core, Department of Pathology, the University of Iowa for processing of fixed tissues and ER and PR staining. We also thank Dr Mikhail Kolonin (UT health science center at Houston) for sharing E0771 cells.

## Author contributions

R.K. and W.Z. conceived and designed the experiments. R.K., Y.L., Q.X., F.L. and W.L. helped obtain data from orthotopic transplant model and staining. N.B., L.P., S.-C.J.Y. and M.-H.L. provided human transcriptome analysis. A.M.J. and R.K. provided *in vitro* inflammasome activation data. M.K. provided materials and helped with the metformin data. M.J.P. and K.R.M. provided differentiated primary mouse adipocytes. L.G.E. provided cell lines and expertise. S.L.C. and F.S.S. provided mice and inflammasome expertise. R.K. and W.Z. analysed data and wrote the manuscript.

## Additional information

**Competing financial interests:** The authors declare no competing financial interests.

