## [Peer Review File · Nature Communications]

Reviewers' comments:

Reviewer #1 (Obesity and Breast Cancer)
(Remarks to the Author):

This is an important and original work addressing the mechanisms of obesity-promoted breast cancer. The authors show that obesity is associated with elevated pro-inflammatory IL-1 signaling in human breast cancer (BC) and in relevant BC mice models. The obesity effects on IL-1 are mediated through NLR4 inflammasome. The experiments also demonstrate that IL-1 β /IL-1R1 axis promotes VEGF expression and angiogenesis. Interestingly, metformin inhibits obesity-associated cancer through the reduction of angiogenesis.

In general, the experiments are robust, well designed and statistically evaluated. The conclusions are appropriate. However, the suggestions of using VEGF inhibitors or metformin as potential treatments of obesity-induced BC have to be taken with caution, as neither of these drugs has shown clear benefit in BC therapy. {Metformin anti-cancer data are derived from diabetes populations only and data in other groups are conflicting, while VEGF medications (Avastin) are not longer approved for BC. Other methods of interfering with obesity pathways should perhaps be considered and addressed in Discussion.

Otherwise, the manuscript is clearly written and I recommend publication.

Reviewer #2 (Inflammation and cancer)
(Remarks to the Author):

Kolb et al demonstrate that high fat diet and obesity promote breast cancer. It was known that obesity predisposes to many types of cancer (including breast) and worsens cancer prognosis, however many previous attempts have concentrated on the direct effects of lipids, insulin and "mTOR-like" signaling in cancer cells. Here, authors demonstrate that hematopoietic cell derived inflammatory circuit constituting of Nlr4 inflammasome and IL-1b acts back on adipocytes to induce excessive VEGF production and enhanced angiogenesis and tumor growth in several mouse models of breast cancer. A good deal of molecular and cellular mechanisms which underlie the effects of obesity- Nlr4, IL-1b, adipocytes, VEGF is further sorted out. Furthermore, key observations are tested via interrogation of large expression data sets of patients with breast cancer, to demonstrate that indeed inferred correlations are quite possible to be true. Overall, this paper is very solidly done, with both significant elements of novelty and enough insights into mechanistic details of how obesity- tumor growth connection work. Several experimental and mechanistic parts of the manuscript however should be further clarified and some additional details for description and discussion need to be provided.

Major critique:

- 1) Since IL-1 is upregulated in tumors and Nlr4 is also expressed there, authors should stain for adipocytes and their precursors in the tumors, if IL-1 indeed acts on adipocytes to induce VEGF.
- 2) Authors should try to at least extensively discuss, whether Nlr4- HFD-IL-1 axis may be in part working because of the distant control of inflammation by microbiota? Are Nlr4 KO mice dysbiotic with or without HFD? That would also help to guess about possible stimuli activating Nlr4 - indeed, so far only bacterial stimuli not present in breast tumors were shown to activate Nlr4 inflammasome. Recent papers from Janelle Ayres and Gabriel Nunez explore some of the systemic roles of Nlr4 and

its connection to microbiota. Authors also need to explicitly state whether WT and KO mice they use are littermates, cagemates, co-housed or separately bred lines housed in different cages etc in order to have a clear picture about their microbiota.

3) It is essential that authors do not bring the readers deeper into the M1/M2 macrophages controversy. Several recent papers and opinions written by the experts in the field that M1/M2 is a useful paradigm for in vitro terminal differentiation of macrophages, which nevertheless is impossible to detect in vivo, at least in a form of stable macrophage lineages with distinct phenotype- see for example PMID: 25035950; PMID: 24812208 and <http://www.cell.com/trends/cancer/abstract/S2405-8033%2815%290007-0-9>. Clearly, F4/80 Cd11b+ Cd11c- are not M2, they are just more mature macrophages while F4/80 Cd11b+ Cd11c+ are closer to monocytes, but later will give rise particularly to F4/80 Cd11b+ Cd11c- macrophages. Arginase 1 and IL-10 indeed could be overexpressed in tumors, but authors do not differentiate between Cd11c+ and Cd11c- cells in terms of IL-10 and arginase expression. I would suggest to re-phrase/re-write that part of the manuscript, because later on the main observation is that myeloid cells produce IL-1 β and IL-1 β stimulates VEGF- there is absolutely no need to chase after M1 and M2 here.

Minor:

- 1) Suppl Fig 1B,C need much better description- what are color codes (green, red), how changes are quantified, some discussion on how pathways work at protein level- provided data is mRNA expression data which is not always translated into protein levels and moreover many proteins in these pathways are regulated by activation, not by differential expression
- 2) Authors should explain better in the text the rationale of using Cont ND diet, what are the differences from HFD and from ND- this is not clear for a regular reader.
- 3) In other models of cancer, IL-6 and TNF were found to be upregulated with HFD and obesity. Authors should discuss that and try to guess in the discussion why in some models IL-6 and TNF are induced, while in breast cancer it is mostly Nlr4-IL-1 pathway.
- 4) MDSC is the same as neutrophil in the context of cancer. More precisely, MDSC are a mixture of monocytes and neutrophils.
- 5) Fig 4A- not clear why some parts of the data are marked with a red frame and what it is supposed to illustrate- please explain?
- 6) For Flow cytometry methods section- both CD11b and CD11c antibodies are listed in Pacblue color, later on double color Cd11b CD11c staining is described- please explain/correct
- 7) Ref#7 is a 1/2 page opinion/description by a Science magazine editor. It is quite unusual to cite that and not some original paper or a comprehensive review.
- 8) How experiments with conditioned BMDM medium (Fig 5) are done and how exactly CM is collected (stimulation, time, dilution), especially- does it still contain IL-1 β used for BMDM stimulation?
- 9) Authors should clearly state what metformin inhibits when they introduce whether its mTOR, mTORC1 or AMPK activator/mTOR inhibitor.
- 10) Authors should cite some previous papers where it was shown that metformin can suppress the expression of various IL-1 dependent genes- *The Journal of Clinical Endocrinology & Metabolism* 92(8):3213-3218; *Arterioscler Thromb Vasc Biol.* 2006;26:611-617

Reviewer #3 (Inflammasomes and diseases)
(Remarks to the Author):

Manuscript #: NCOMMS-16-02466-T

Manuscript Title: Obesity-associated NLRC4 inflammasome activation drives breast cancer progression

Authors: Ryan Kolb, Liem Phan, Nicholas Borchering, Yinghong Liu, Fang Yuan, Ann M Janowski Qing Xie, Kathleen R Markan, Wei Li, Matthew J Potthoff, Enrique Fuentes-Mattei, Lesley G Ellies, Michael Knudson, Mong-Hong Lee, Sai-Ching Jim Yeung, Suzanne L Cassel, Fayyaz S. Sutterwala, and Weizhou Zhang

In this manuscript, Kolb et al characterize a possible role of NLRC4, in an obesity induced breast cancer via an inflammasome dependent manner. The authors found that the combination of a high fat diet plus the absence of NLRC4 resulted in more IL-1 β and enhanced growth of breast cancer tumors. The production of IL-1 was dependent on the inflammasome component caspase-1. The production of IL-1 β via tumor-infiltrating myeloid cells drives disease progression through adipocyte-mediated VEGFA expression. Additional analysis of different databases including Geo Dataset and TCGA revealed a correlation with obese breast tissue and increased levels of NLRC4 expression. Analysis of TCGA database revealed increase levels of CASP1 expression in tumors from overweight patients and high levels of NLRC4 was associated with invasive ductal carcinoma data. While no mechanism in how Nlrc4 is activated is determined, the authors present interesting observations linking NLRC4 to obesity associated tumor progression. However there are several areas of concerns that have to be addressed.

1. Values that are of statistical significance should be indicated in all of the figures. In some figures, it is not marked what the statistical value refers to (e.g., figure. 1g).
2. It is difficult to understand why IL-1 β deficient mice do not have obese phenotype but caspase-1 knockout mice do. The caspase-1 knockout mice should not produce IL-1 β . Could this be due to the confounding impact of IL-18 loss which causes over eating? Have the authors any data on basic metabolic measurements, such as food intake and activity?
3. In Supplemental figure 1, only "pathways" are indicated, without the data of gene expression differences in patients or mice. For this data to be transparent and useful, more information is needed. Most important, how are the different inflammasomes genes altered in both (IL-1b, IL-18, ASC, caspase 1, caspase 11, NLRP3 and NLRC4 to mention a few.
4. Some clarification is needed for the caspase-1 knockout. Is this the caspase-1/caspase-11 deletion or caspase-1 deletion strain? If it is a caspase1/11 then a delineation of these two genes in their model should be included.
5. There are published reports that discuss the correlation between increase protein and serum levels of VEGF and certain cancers (i.e. colorectal cancer and lung cancer). The authors only address the mRNA levels of Vegfa family members. Is there a difference in VEGF in the serum of the mice or is there only a message difference increase found in the tumors? Does the HFD diet induce serum VEGF? Once metformin is added, is there a difference in VEGF serum levels?
6. Figure 2 a, the filled symbols do not match the legend. For example, there is a gray square in the figure, but the legend does not have a gray square. Figure 2f: E0771 tumor volumes are typical 2 fold more then compared to Py8119 cells. Is this due to the difference in ER status of cells? Or the origin of the cell type?
7. On page 3: The abbreviations of IDC and DCIS are used for the first time, please define what they are.
8. In Figure 3e: What is activating Nlrc4 expression in CD11b+ cells? As a simple experiment, would any component in a HFD induce NLRC4 expression in macrophages in vitro?

9. Figure 4a: The authors mention a difference in chemokines between HFD & ND mice in the result section but Figure 4a is unlabeled on the left or right side so the reader can't determine what they are looking at. Plus there are red boxes on the figure and there is no explanation for it.

10. Figure 4b: this is very difficult to interpret. Are there any negative controls for staining. It is not clear what is stained since there are no biomarkers. Same with Fig. 6c. Arrows would be helpful in determining what cells are CD31 positive. Need to add label of Py8119 to Figure 4e

11. In Figure 5d, the authors need to show some direct evidence that JNK is activated in the adipocytes from HFD fed mice. Only inhibitor strategy is not convincing enough. Either immunostaining or Western blots is needed.

12. In supplementary figure 4c-d, the authors notice a difference in IL-18mRNA levels in the two cell type models, with Py8119 tumor cells has more IL-18 compared to E0771. Could this be due to the difference in cell type and the ER status of the cells? Could the authors include some explanation as to why there is such a difference?

13. Figure 6F: Does the application of metformin affect the production of pro-angiogenic inflammatory chemokine's CXCL1, CCL2, and CCL3?

14. Figure 7e: Analysis of TCGA invasive ductal carcinoma data showed higher NLRC4 expression and was associated with shorter survival. Analysis of another NLR such as NLRP3 which showed a null effect in their animal study should serve as a negative control. What is the level of Vegf expression in this tissue?

Reviewer 1:

Comments:

..... the suggestions of using VEGF inhibitors or metformin as potential treatments of obesity-induced BC have to be taken with caution, as neither of these drugs has shown clear benefit in BC therapy. {Metformin anti-cancer data are derived from diabetes populations only and data in other groups are conflicting, while VEGF medications (Avastin) are not longer approved for BC. Other methods of interfering with obesity pathways should perhaps be considered and addressed in Discussion.

Otherwise, the manuscript is clearly written and I recommend publication.

Response

Thank you for the positive comments and the recommendation for publication.

We appreciate the point that Avastin has failed for breast cancer therapy. We have removed the suggested therapeutic benefit of anti-angiogenesis and metformin in the abstract in respect of this comment. In the abstract we now focus on the potential use of targeting the inflammasome or IL-1 signal transduction. We also discussed the potentially beneficial effects of anti-angiogenesis therapies in highly angiogenic cancer patients, as suggested by a recent PNAS paper (reference 47, *Proc Natl Acad Sci U S A* **112**, 14325-14330 (2015)). The trial for metformin in obese breast cancer patients is ongoing. Until the final result, we won't know for sure if it will be beneficial. Considering its safety and its effects on body weight and potentially longevity, metformin holds promise for improving overall health especially for obese individuals. Related to cancer therapy, we may need to identify the specific patient population that will benefit from metformin treatment, such as obese patients with high angiogenic cancer.

Reviewer 2:

Thank you for your remarks and insightful critiques of the submitted manuscript. We appreciate your suggestions that will help to clarify some of the experimental procedures and discussions. Below are the responses to your specific critiques:

Major Critiques:

- 1) "Since IL-1 is upregulated in tumors and NLRC4 is also expressed there, authors should stain for adipocytes"

Response:

We have shown that NLRC4 is primarily expressed by tumor infiltrating macrophages that are the source of IL-1 β . IL-1 β consequently works 1) directly through adipocytes in the obese tumor microenvironment or 2) through macrophages to turn on some unknown factors to induce Vegfa expression from adipocytes. The increased angiogenesis and tumor growth in obese animals are driven by the signaling crosstalk between adipocytes and macrophages. We stained tumor sections from ND and HFD mice with Oil-O red stain, which stains for triglycerides and lipids and is a good marker for adipocytes. We observed a general increase in adiposity in tumors from HFD mice compared to ND mice, suggestive of the increased number of adipocytes (New Supplementary Fig 5f).

- 2) "Authors should try to at least extensively discuss whether Nlrc4-HFD-IL-1 axis may be in part working because of the distant control of inflammation by microbiota? Are Nlrc4 mice dysbiotic with or without HFD?" "Authors also need to explicitly state with WT and KO mice were littermates, cage mates, co-housed or separately bred lines housed in different cages etc. in order to have a clear picture of the microbiota"

Response:

We appreciate the comments and suggestions to discuss the possible changes in the microbiota due to loss of NLRC4 and/or HFD treatment. NLRC4 knockout mice do have changes in gut microbiota compared to WT mice, as shown by Ellinav E, et al. *Cell* 2011. Feeding mice a HFD also leads to changes in the gut microbiota compared to mice given a ND. HFD also leads to increased endotoxemia, which plays a major role in HFD-induced inflammation, obesity and metabolic syndrome in mice (Cani, P.D. et al. *Diabetes* 2008). As such, the HFD-Nlrc4-IL-1 axis may work in part through HFD-induced changes in the gut microbiota and endotoxemia-induced inflammation. All knockout mice were extensively backcrossed to C57BL/6N to achieve at least 97% genetic identity; whereas we used the same C57BL/6N female mice as wild type controls. We confirmed NLRC4 and CASP1 knockout mice with littermates and others using age-matched wild type C57BL/6N females purchased from NCI and later from Charles River since NCI terminated the animal stocks. For obvious reasons, ND- and HFD-

fed mice could not be co-housed. To minimize microbiota change, we mixed dirty bedding from all cages once a week and added the mixed dirty beddings into the clean bedding of each cage. This was further discussed in the revised manuscript in the 2nd paragraph of the discussion where we discuss possible mechanisms for obesity-induced Nlrc4 activation. The methods section was also revised accordingly.

3) “It is essential that authors do not bring the readers deeper into the M1/M2 macrophage controversy”

Response:

We appreciate the comments on the distinction between M1 and M2 macrophages and the suggestion to not get involved in this controversy. We agree that the M1 and M2 macrophage model is based on in vitro experiments and likely represents opposite extremes of a spectrum of macrophage polarization, with most macrophages existing somewhere in between the two in vivo. As suggested, we revised that section of the manuscript and refrained from mentioning M1 and M2 macrophages. We changed Figure 3a to CD11c⁺ and CD11c⁻ macrophages instead of M1 and M2. We also purposely removed all M1/M2 polarization-related contents.

Minor critiques:

1) “Suppl fig 1B,C need much better description-“ “some discussion of how pathways work at a protein level”

Response:

In Supplementary Figure 1b and 1c, we wanted to see if obesity was associated with changes in the gene expression of the IL-1 pathway, with the protein data nearly impossible to get at this time. We included expression data from both human breast cancer patients (b) and an animal model (c) in which obesity promoted disease progression. Using Ingenuity Pathway Analysis software, we compared obese patients and mice to non-obese patients and generated z-scores for the genes in the IL-1 pathway. The Z-scores were represented as a color, with red being a positive z-score, meaning that there is a positive correlation between the expression of that gene and obesity, and green being a negative z-score (new Supplementary Figure 1d). These gene expressions were overlaid on the IL-1 signaling pathway at the protein level to reflect the transcriptional increase at pathway level. A heat map of these Z-scores for individual gene expression was added to Supplementary Figure 1d to better clarify the Figure using a Green-Black-Red scale.

2) “Authors should explain better in the text the rationale for using Cont ND diet, what are the differences from HFD and from ND- this is not clear for the regular reader.”

Response:

We appreciate the comment on clarifying the use of the Cont ND. The control ND is the control diet from BioServ, specifically designed for controlling the macronutrients for its HFD. It is composed of the same macronutrients as the HFD, except the HFD diet is 36% fat and 35.7% carbohydrates, while the Cont ND is 7.2% fat and 61.6% carbohydrates. We used this as a control to ensure that the differences we observed in tumor progression were due to the obesity caused by the increased fat intake and not some other change in the macronutrients. This was discussed further in the revised manuscript under the Methods section.

3) “In other models of cancer IL-6 and TNF were found to be upregulated with HFD and obesity. Authors should discuss that and try to guess in the discussion why in some models IL-6 and TNF are induced while in breast cancer it is mostly Nlrc4-IL-1 pathway.”

Response:

IL-6 and TNF α have been implicated to play a role in obesity associated cancers in various tissues. In breast cancer, most of the studies are in vitro and there is no causal relation between IL-6 or TNF α and breast cancer progression under obesity. We do not fully understand why obesity in breast cancer preferentially signals through the NLR4-IL-1 pathway rather than the IL-6 and TNF α pathways in the mouse models used and in human breast cancer. However, a recent study examining the role of IL-6 and TNF genes in modulating breast cancer risk failed to find strong evidence linking them to breast cancer risk. (Margaret M. et al. Breast Cancer Res Treat. 2011 Oct; 129(3): 887–899.). Furthermore, no GWAS of breast cancer have generated evidence of breast cancer risk associate with variants in IL-6 or TNF α .

4) “MDSC is the same as neutrophil in the context of cancer”

Response:

We revised that section of the manuscript to label those cell populations as granulocytic MDSC/neutrophils.

- 5) "Fig 4A- not clear why some parts of the data are marked with red frame and what it is supposed to illustrate- please explain"

Response:

The red frames are present to distinguish the reference spots to show equal loading between the two blots. This was clarified in the figure legends and text of the revised manuscript.

- 6) "For flow cytometry methods section- both CD11b and CD11c antibodies are listed in PacBlue color"

Response:

The CD11b antibody was FITC color while the CD11c was PacBlue. This mistake was corrected in the revised manuscript.

- 7) "Ref #7 is a ½ page opinion/description by a science magazine editor. It is quite unusual to cite that and not some original paper or a comprehensive review."

Response:

The reference was changed in the revised manuscript.

- 8) "How experiments with conditioned BMDM medium (fig 5) are performed and how exactly CM is collected"

Response:

This was clarified in the methods of the revised manuscript. Briefly, BMDM were treated with 100ng/ml of IL-1 β for 72 hrs and then CM was collected. The CM still likely contains some IL-1 β , however the CM induced much greater expression of *Vegfa* in adipocytes compared to treatment with the same concentration of IL-1 β used to treat the macrophages.

- 9) "Authors should clearly state what metformin inhibits when they introduce"

Response:

In the revised manuscript, we added a brief discussion of what metformin inhibits. Metformin inhibits mitochondrial complex 1 of the respiratory chain leading to AMPK activation. It is unclear whether metformin inhibits obesity-associated angiogenesis and tumor growth through this mechanism or some other mechanism.

- 10) "Authors should cite some previous papers where it was shown that metformin can suppress the expression of various IL-1 dependent genes"

Response:

We referenced the suggested papers and added them to the first paragraph of the discussion.

Reviewer 3:

Thank you for your comments and helpful critiques on the submitted manuscript. Here we will address the specific concerns that you had:

- 1) "Values that are significance should be indicated in all of the figures"

Response:

We have marked all significant values in the figures of the revised manuscript and clearly stated what the values refer to in the figure legends.

- 2) "It is difficult to understand why IL-1 β mice do not have obese phenotype but Casp-1 knockout mice do" "Could this be due to the confounding impact of IL-18 loss which caused over eating?" "Have the authors collect any data on basic metabolic measurements such as food intake and activity?"

Response:

We appreciate the comment on the discrepancy between IL-1 β mice and Casp1 mice in regards to the obesity phenotype. This has been known in the IL-1 β and inflammasome field, while the mechanism of the distinction is unknown. It is likely in part due to the concomitant loss of IL-18 and IL-1 β in Casp1 mice. Previous studies have shown that loss of IL-18 leads to hyperphagia, obesity and insulin resistance (Netea, M.G. et al. Nat Med. 2006). We did not house individual mouse in metabolic cages so we do not have any activity measurements. We did measure the amount of food the mice in each cage

ate every week by weighing the food. All of the mice on the HFD ate less food by weight, however the food has a higher caloric density compared to ND (5.49 kcal/g vs 3.93 kcal/g).

- 3) “ In supplemental figure 1, only pathways are indicated, without the data of gene expression differences in patients or mice. For this data to be transparent and useful, more information is needed. Most important, how are the different inflammasome genes altered in both”

Response:

In the revised manuscript, we have added a heat map for the differentially regulated genes between obese patients and mice in the pathways in Supplemental Figure 1d. The Affymetrix chip used to generate this expression data set did not contain a probe for NLRC4, so we cannot include the NLRC4 expression data.

- 4) “Some clarification is needed for the Caspase-1 knockout. Is this the Casp1/Casp11 deletion or Casp1 deletion strain?”

Response:

The mice used in the manuscript are the Casp1/Casp11 deletion strain and the reference is cited in the revised manuscript (reference 23 in the revised manuscript). As such, to address the concern about the roles of these two genes in our model, we looked at Casp11 processing by western blot in CD11b+ and CD11b- population from ND and HFD tumors. We found that Casp11 was primarily in CD11b+ myeloid population. We also found that Casp11 was not activated by HFD (as evidence by presence of the cleaved form p18). In fact, Casp11 activation was slightly lower in tumors from HFD mice. This suggests that Casp11 activation should not play a role in our model. However, without the use of Casp11 knockout mice, we cannot completely rule out that possibility, Therefore, we changed to Casp1/Casp11 knockout instead of Casp1 knockout. The Casp11 processing data was added into Fig.3c in the revised manuscript.

- 5) “The authors only address the mRNA levels of Vegfa family members. Is there a difference in VEGF serum of the mice or is there only a message difference found in the tumors?” Does the HFD induce serum VEGF” Once metformin is added is there a difference in VEGF serum levels”

Response:

We have tried to collect serum for the experiments based on the suggestion, but failed since we cannot reliably detect VEGFA serum level in our system. We also did not see the protein level in the proteome array for angiogenesis panel (Fig. 4a). There are many reports showing that obesity increase serum VEGF levels and angiogenesis in mouse (*J Nutr Biochem.* 2010 Aug;21(8):774-80 from Fruhbech group and several other reports) and human (*Plos One.* 2010. Sep. 5(9):e12610 from Oltmanns KM lab and several others). Previous studies have shown that metformin can reduce serum VEGF levels in both diabetic and non-diabetic mice in a HFD and streptozotocin induced diabetic model (Zaafar, D.K. et al. *PLoS One* 2014). Furthermore, other studies have shown that metformin can reduce serum VEGF in diabetic patients (Ersoy, C. et al. *Diabetes res clin pract* 2008). Since the effect of metformin on serum VEGF is already known, we included these citations to our manuscripts to support a protein level change of VEGFA. All these references were included in the revised manuscripts.

- 6) “Fig 2a, the filled symbols do not match the legend.” “ Figure 2f: E0771 tumor volumes are typical 2 fold more then compared to Py8119 cells. Is this due to the difference in ER status of cells? Or the origin of cell type”

Response:

Figure 2a was corrected in the revised manuscript.

E0771 tumors do grow faster than Py8119 tumors. It is unclear why these tumors grow faster. Most cancer cells have different tumor kinetics due to their intrinsic difference. Py8119 cells were isolated from a tumor that arose in an MMTV-PyMT mouse, while E0771 cells were derived from an adenocarcinoma of the breast that developed spontaneously in a C57BL/6 mouse. Py8119 cells are ER- while E0771 cells are ER+. The difference in tumor growth could from any number of reasons, and would be impossible for us speculate on the reason without many experiments.

- 7) “On page 3: the abbreviates of IDC and DCIS are used for the first time please define what they are”

Response:

This was defined in the revised manuscript.

- 8) “ In figure 3e what is activating Nlrc4 expression in CD11b+ cells? As a simple experiment would any component in a HFD induce NLRC4 expression in macrophages in vitro?”

Response:

We found NLRC4 is induced by obesity in mammary tumors (Fig. 3e) as well as in the breast from obese human (Fig. 7a-7b). Thus the increase of NLRC4 expression in myeloid cells in an obese microenvironment appears to be conserved in mouse and human. Obesity increases the number of total infiltrating macrophages that are the source of NLRC4 expression. At a cellular level, we don't know the cause of NLRC4 elevation and there are too many potential causes such as 1) the metabolic changes associated with obesity, including the increase in free fatty acids, triglycerides and their metabolic derivatives (such as oxidized fatty acids and lipids), 2) the positive feedback from activation of inflammasome/IL-1 β and downstream NF- κ B activation, 3) the changes in inflammatory signaling lipids such as eicosanoids, or 4) a combination of any of these and other mechanisms such as circulating bacterial products. This complexity is impossible to recapitulate in vitro.

Practically, the HFD is essentially just a piece of fat pellet and cannot be used to treat cells for NLRC4 activity. We did purify lipids from HFD tumors versus ND tumors and failed to activate NLRC4 in either case (data not shown).

- 9) "Fig 4a: The authors mention a difference in chemokines between HFD and ND mice in the result section but figure 4a is unlabeled on the left or right side so the reader can't determine what they are looking at. Plus there are red boxes on the figure and there is no explanation for it."

Response:

Figure 4a is an angiogenesis profiler array. The left panel is a membrane dotted with specific antibodies for various proteins associated with angiogenesis in pairs. The membranes were incubated with lysates of tumors from either ND or HFD mice. Each pair of dots represents a blot for 1 single protein. The right panel is an analysis of the proteome profiler array. The mean pixel density of each spot was determined using image J software, and normalized for protein loading based on the mean pixel density of the reference spots (the red boxes). The mean pixel density of select proteins that were differentially regulated was then presented as a fold change compared to ND. The proteins that each bar represents are labeled on the x-axis of the graph on the right. The red boxes represent the reference spots to show equal loading. This information was added to the figure legend of the revised manuscript. We cannot make too many labels on the two blots since so many proteins were found upregulated.

- 10) "Fig 4b: this is very difficult to interpret. Are there any negative controls?" "Arrows would be helpful in determining what cells are CD31 positive. Need to add label of Py8119 o figure 4e."

Response:

This is a classic immunohistochemistry to show specific antigen staining, with the brown color labeling specific CD31 staining and hematoxylin to label nuclei. We added a few white arrows pointing to the brown color indicative of endothelial cells. A negative isotype control was added to Figure 4b.

The label of Py8119 was added to figure 4e.

- 11) "In figure 5d, the authors should show some direct evidence that JNK is activated in adipocytes from HFD-fed mice."

Response:

We treated primary mammary adipocytes with 100 ng/ml IL-1 β for 30, 50, and 120 mins and looked for JNK activation by immunoblotting for pJNK. We found that IL-1 β leads to JNK phosphorylation in within 30 mins, and is decreased after 2 hours. These data support our finding that IL-1 β induced *Vegfa* expression in adipocytes through JNK activation. These data were added to Figure 5e in the revised manuscript.

- 12) "The authors notice a difference in IL-18 mRNA levels in the two cell type models." "could this be due to the difference in cell type and ER status of the cells? Could the authors include some explanation why there is such a difference."

Response:

We did observe an increase in IL-18 mRNA in Py8119 tumors in mice fed a HFD but not in E0771 tumors fed a HFD. It is unclear why one model had increased IL-18 expression and not the other. It may be due to difference in cancer cell types that elicit the different composition of myeloid cells, which results in different expression levels of IL-18.

- 13) "Does the application of metformin affect the production of pro-angiogenic inflammatory cytokines CXCL1, CCL2, CCL3?"

Response:

We looked at the expression of CXCL1, CCL2, and CCL3 in tumors from ND, HFD, *NLR4*^{-/-} HFD, and HFD + metformin mice. CXCL1 is upregulated in the HFD mice, but its expression is not regulated by *NLR4* inflammasome activation or metformin treatment. CCL2 and CCL3 mRNA was not upregulated by the HFD. CCL2 expression, however, was increased by treatment with metformin. From these data we believe that Vegfa is the primary factor that is impacted by the metformin and *NLR4*/IL-1 β pathways in obesity. These data were added to the revised manuscript in Supplementary Figure 5e.

- 14). “ Figure 7e: analysis of TCGA invasive ductal carcinoma data showed higher *NLR4* expression and was associated with shorter overall survival. Analysis of another NLR such as *NLRP3* which showed a null effect in their animal study should serve as a negative control. What is the level of Vegf expression in this tissue?”

Response:

We repeated the analysis of the association between *NLR4* or *NLRP3* expression and overall survival using the updated TCGA data. We divided the data into tertiles based on survival and compared high versus low expression groups. *NLR4* expression is associated with shorter overall survival, while *NLRP3* expression has no correlation with survival. We also repeated the analysis using KM Plot which normalizes expression across several breast cancer datasets. We found that high *NLR4* expression was associated with shorter survival overall in all breast cancer samples (HR=1.49, p=0.051) and ER+ breast cancers (HR=1.88, p=0.0072). In contrast, *NLRP3* expression was correlated with longer overall survival in ER+ breast cancers all breast cancers. These new data were added to the revised manuscript in Figure 7e, and the TCGA data were moved to a supplementary Figure 7a-7b. We also checked to see if there was a correlation between *NLR4* expression and VEGFA in the TCGA dataset but found no correlation.

REVIEWERS' COMMENTS:

Reviewer #2 (Remarks to the Author):

Authors did a very good job addressing reviewer's criticism and I do not have any burning major criticism and think that the paper now can be accepted. I think observation that in typically non-inflammatory breast cancer obesity can drive tumor progression via inflammatory mechanisms is an important, novel and long awaited observation.

Reviewer #3 (Remarks to the Author):

The authors have address most of the concerns and the work has substantial new information. Thus I favor acceptance.

We appreciate both reviewers' comments and the acceptance of our manuscript. No further comment is raised by either reviewer.